# Scalable Generation of Spatial Transcriptomics from Histology Images via Whole-Slide Flow Matching

**Tinglin Huang** [1]  **Tianyu Liu** [1]  **Mehrtash Babadi** [2]  **Wengong Jin** [2 3]  **Rex Ying** [1]

## Abstract

Spatial transcriptomics (ST) has emerged as a powerful technology for bridging histology imaging with gene expression profiling. However, its application has been limited by low throughput and the need for specialized experimental facilities. Prior works sought to predict ST from whole-slide histology images to accelerate this process, but they suffer from two major limitations. First, they do not explicitly model cell-cell interaction as they factorize the joint distribution of whole-slide ST data and predict the gene expression of each spot independently. Second, their encoders struggle with memory constraints due to the large number of spots (often exceeding 10,000) in typical ST datasets. Herein, we propose STFlow, a flow matching generative model that considers cell-cell interaction by modeling the joint distribution of gene expression of an entire slide. It also employs an efficient slide-level encoder with local spatial attention, enabling whole-slide processing without excessive memory overhead. On the recently curated HEST-1k and STImage-1K4M benchmarks, STFlow substantially outperforms state-of-the-art baselines and achieves over 18% relative improvements over the pathology foundation models.

## 1. Introduction

Compared to the early days of bulk RNA sequencing, recent advancements in spatial transcriptomics (ST) technology offer a novel approach to molecular profiling within the spatial context of tissues, providing insights into cellular interactions and the microenvironment (Ståhl et al., 2016; Xiao & Yu, 2021). One of the promising clinical applications of ST is the prediction of biomarkers in digital pathology, often visualized in hematoxylin and eosin (H&E)–stained whole-slide images (WSIs), by analyzing the gene expression levels in relation to the tissue morphology (Levy-Jurgenson et al., 2020; Zhang et al., 2022). However, the conventional ST methods (Moffitt et al., 2018; Eng et al., 2019; Ståhl et al., 2016) suffer from low throughput and need of specialized equipment, limiting their application compared to standard histology imaging.

To address this, recent works resort to deep learning to predict spatially-resolved gene expression from H&E images. As illustrated in Figure 1(a), a histology image is segmented into small spots, with the objective of predicting the gene expression with the spot image and the coordinate. This line of research has achieved promising results using an image foundation model to encode local spot-level features (Chen et al., 2024b; He et al., 2020; Ciga et al., 2022; He et al., 2016), but they neglect the utilization of spatial dependencies between spots. To bridge this gap, some studies introduce an additional slide-level encoder to incorporate global spatial context (Xu et al., 2024; Chung et al., 2024; Pang et al., 2021; Zeng et al., 2021) with an vision transformer.

Despite their initial success, these methods are limited by (1) high computational complexity: the exhaustive attention mechanism among all spots leads to significant computational overhead, making it impractical for standard gigapixel slides, which contain tens of thousands of spots (Campanella et al., 2019; Lu et al., 2021); (2) weak utilization of spatial dependencies: these methods typically encode coordinates as positional embeddings, making them sensitive to numerical noise and variations in coordinate scales caused by batch effects. (3) overlooking cell-cell interaction, i.e., certain genes regulating the expression of genes in other cells (Figure 1(b)) (Li et al., 2022; Biancalani et al., 2021).

In light of this, we propose STFlow[1], a flow matching-based model that reformulates the original regression task as a generative modeling problem. Instead of performing one-step regression, we model the joint distribution over the whole-slide gene expression through an iterative refinement

---

[1]Yale University. [2]Broad Institute of MIT and Harvard. [3]Northeastern University. Correspondence to: Tinglin Huang <tinglin.huang@yale.edu>.

*Proceedings of the 42^{nd} International Conference on Machine Learning*, Vancouver, Canada. PMLR 267, 2025. Copyright 2025 by the author(s).

[1]Implementation can be found at https://github.com/Graph-and-Geometric-Learning/STFlow

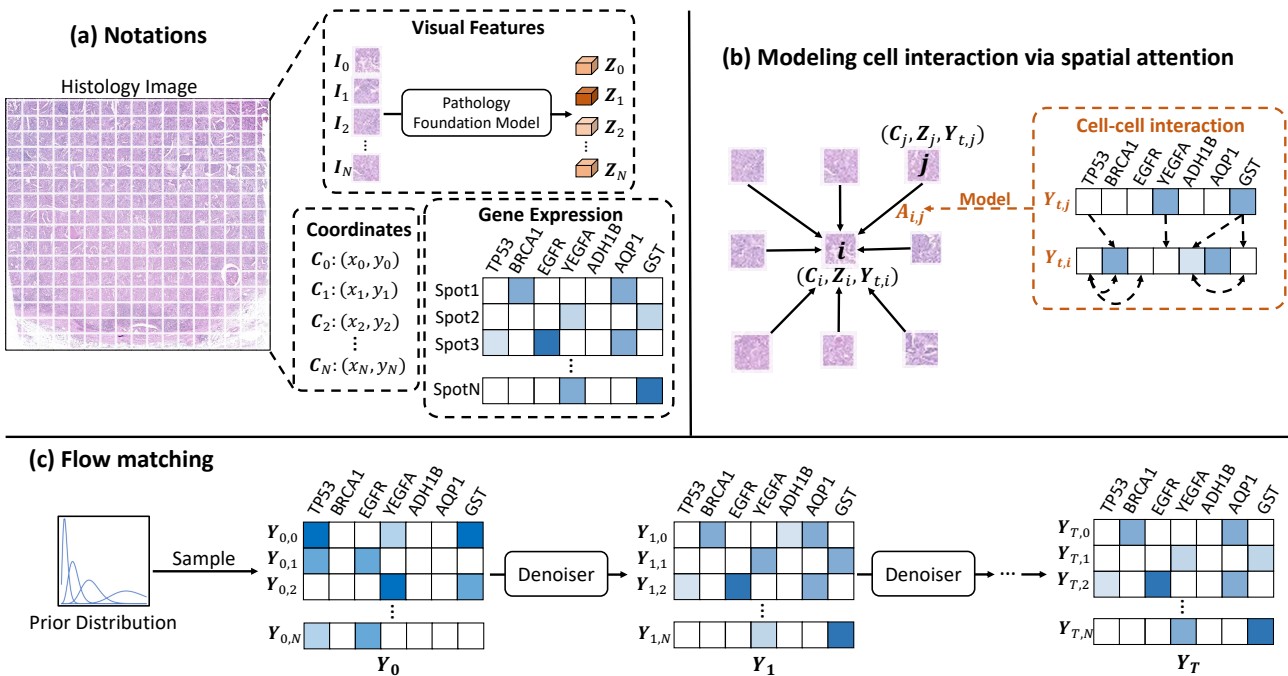

*Figure 1.* An overview of gene expression prediction from histology image with STFlow. **(a)**: The histology image is segmented into a set of spot images, each associated with a 2D coordinate and gene expression. Each spot image is then encoded using a pathology foundation model. **(b)**: STFlow encodes the slide-level context by aggregating the neighboring spots through spatial attention and the interaction between cells is explicitly incorporated within the attention calculation. **(c)**: STFlow iteratively optimizes the gene expression predictions, starting from a sample drawn from a prior distribution, such as zero-inflated negative binomial (ZINB) distribution.

process, as depicted in Figure 1(c). Each refinement step is guided by the flow matching framework, where the predicted gene expression serves as context for the subsequent step. This enables explicit modeling of cell-cell interactions, leading to more biologically meaningful predictions. The denoising network employs a frame averaging (FA)-based (Puny et al., 2021) spatial model with E(2)-invariant spatial attention and achieves great efficiency by modeling the local context of each spot.

We evaluate STFlow on HEST-1k (Jaume et al., 2024) and STImage-1K4M (Chen et al., 2024a), two large-scale ST-WSI collections comprising a total of 17 benchmark datasets. Compared to five spot-based and three slide-based methods, STFlow consistently outperforms all baselines and achieves an 18% average relative improvement over the pathology foundation models. It also excels in the prediction of 4 biomarker genes, highlighting its clinical potential. Moreover, our proposed architecture offers orders-of-magnitude faster runtime and lower memory cost than existing slide-based approaches.

## 2. Related Work

**WSI-based spatial gene expression prediction** Rapid advances in spatial transcriptomics (ST) (Li & Wang, 2021)

have enabled the detection of RNA transcript spatial distribution at sub-cellular resolution (Moffitt et al., 2018; Codeluppi et al., 2018; Eng et al., 2019; Ståhl et al., 2016; Stickels et al., 2021). Machine learning-based approaches have recently shown promising results in predicting expression from histology image (Lee et al., 2023). The previous studies fall into two categories: **(1) spot-based approaches** which solely encode the spot and predict the gene expression individually, i.e., modeling $p(\boldsymbol{Y}_i|\boldsymbol{I}_i)$ (He et al., 2020; Pang et al., 2021; Chen et al., 2024b; Ciga et al., 2022; Xie et al., 2023). Some of these methods leverage foundation models pretrained on large-scale digital pathology datasets, achieving promising results in gene expression prediction (Jaume et al., 2024). One concurrent work (Zhu et al., 2025) leverages diffusion models for ST gene expression generation, but they still treat each spot independently. In contrast, our generative model operates at the whole-slide level, explicitly modeling the joint distribution of genes across spots. **(2) slide-based approaches** which incorporate the slide-level context and predict the gene expression of each spot individually, i.e., modeling $p(\boldsymbol{Y}_i|\boldsymbol{I}_0, \cdots, \boldsymbol{I}_N)$ (Pang et al., 2021; Zeng et al., 2021; Jia et al., 2024; Xu et al., 2024; Chung et al., 2024). The main idea of these methods is to aggregate the representations of other spots after the image encoders extract each spot's features. The key difference between our proposed STFlow and previous methods is that STFlow

explicitly utilizes gene-gene dependency between cells for prediction using a generative model, i.e., modeling the joint distribution $p(\boldsymbol{Y}_0, \cdots, \boldsymbol{Y}_N | \boldsymbol{I}_0, \cdots, \boldsymbol{I}_N)$.

**Flow matching** Flow matching is a generative modeling paradigm (Lipman et al., 2022; Albergo & Vanden-Eijnden, 2022; Liu et al., 2022; Jing et al., 2024; Nori & Jin, 2024) that has shown impressive results across various modalities, including images and biomolecules. The objective is to approximate the marginal vector field of the time-dependent probability path using a neural network. In this work, we reformulate the gene expression regression as a generative task and apply the flow matching since (1) its iterative denoising scheme allows us to incorporate the gene expression within the modeling, and (2) it offers flexibility in selecting a gene expression-specific prior distribution, e.g., zero-inflated negative binomial distribution.

**Geometric deep learning** Geometric deep learning has achieved significant success in chemistry, physics, and biology (Bronstein et al., 2021; Zhang et al., 2023; Liu et al., 2023). The key to this success lies in generating invariant representations for 3D structures, such as molecule conformation, that remain consistent under E($n$) transformations, where $n$ represents the dimension of the Euclidean space. E($n$) transformations include translations, rotations, and reflections. Previous methods achieve invariance by leveraging invariant features (Satorras et al., 2021; Schütt et al., 2018; Gasteiger et al., 2021) or employing equivariant transformations, such as irreducible representations (Fuchs et al., 2020; Liao & Smidt, 2022; Weiler & Cesa, 2019) and frame averaging (FA) (Puny et al., 2021; Huang et al., 2024). The architecture of the denoiser encodes the spatial context of WSIs using an FA-based Transformer architecture, designed to produce invariant representations for each spot, regardless of any E(2) transformations.

## 3. Method

In this section, we introduce STFlow, a flow matching framework for modeling the joint distribution of gene expression across spots, along with an E(2)-invariant denoiser for capturing spatial dependencies. We first introduce the necessary background in Section 3.1 and elaborate on the learning framework in Section 3.2. The introduction of architecture is provided in Section 3.3.

### 3.1. Preliminaries

**Problem Formulation** An H&E-stained WSI is segmented into a set of patches, which can be represented as $(\boldsymbol{C}, \boldsymbol{I}, \boldsymbol{Y})$, with coordinates $\boldsymbol{C} \in \mathbb{R}^{N \times 2}$, spot images $\boldsymbol{I} \in \mathbb{R}^{N \times 3 \times H \times W}$, and gene expression levels $\boldsymbol{Y} \in \mathbb{R}^{N \times G}$, where $N$ is the number of spots, $G$ is the number of genes, and $H, W$ indicate the image dimensions. Each element in

---

**Algorithm 1** STFlow: Training

**Require:** Training WSIs $(\boldsymbol{C}, \boldsymbol{I}, \boldsymbol{Y})$
Sample prior $\boldsymbol{Y}_0 \sim \mathcal{Z}(\mu, \phi, \pi)$
Sample timestep $t \sim \text{Uniform}[0, 1]$
Interpolate $\boldsymbol{Y}_t \leftarrow t \cdot \boldsymbol{Y} + (1 - t) \cdot \boldsymbol{Y}_0$
Predict $\hat{\boldsymbol{Y}} \leftarrow f_\theta(\boldsymbol{C}, \boldsymbol{I}, \boldsymbol{Y}_t, t)$
Minimize objective $\text{MSE}(\boldsymbol{Y}, \hat{\boldsymbol{Y}})$

---

**Algorithm 2** STFlow: Inference

**Require:** Testing WSIs $(\boldsymbol{C}, \boldsymbol{I})$
Sample prior $\boldsymbol{Y}_0 \sim \mathcal{Z}(\mu, \phi, \pi)$
**for** $s \leftarrow 0$ **to** $S - 1$
    Let $t_1 \leftarrow s/S$ and $t_2 \leftarrow (s + 1)/S$
    Predict $\hat{\boldsymbol{Y}} \leftarrow f_\theta(\boldsymbol{C}, \boldsymbol{I}, \boldsymbol{Y}_{t_1}, t_1)$
    **if** $s = S - 1$ **then**
        **return** $\hat{\boldsymbol{Y}}$
    **end if**
    Interpolate $\boldsymbol{Y}_{t_2} \leftarrow \boldsymbol{Y}_{t_1} + \frac{(\hat{\boldsymbol{Y}} - \boldsymbol{Y}_{t_1})}{(1 - t_1)} * (t_2 - t_1)$
**end for**

---

$\boldsymbol{Y}$ is the count of detected RNA transcripts for a particular gene (starting from 0), representing the gene's expression level. In this study, the goal of STFlow is to predict the gene expression $\boldsymbol{Y}$ among spots with the input of $(\boldsymbol{C}, \boldsymbol{I})$.

**Pathology Foundation Model** We define $f_{\text{PFM}}(\cdot)$ as a pathology foundation model, which aims to extract general-purpose embeddings for digital pathology after being pre-trained on large-scale histology slides (Ciga et al., 2022; Chen et al., 2024b; Xu et al., 2024). They receive a patch from the slide as input and produce the embedding for downstream tasks:

$$\{\boldsymbol{Z}_0, \cdots, \boldsymbol{Z}_N\} = f_{\text{PFM}}(\{\boldsymbol{I}_0, \cdots, \boldsymbol{I}_N\}) \qquad (1)$$

where $\boldsymbol{Z}_i, \boldsymbol{I}_i$ represent the $i$-th spot's encoded representation and H&E image.

In our study, we leverage these foundation models to extract visual features for each spot image instead of training an individual image encoder. The motivation is that, after being pretrained on large-scale histology slides, these foundation models exhibit strong generalization abilities and help mitigate batch effect (Jaume et al., 2024).

### 3.2. Learning with Flow Matching

Modeling cell-cell interaction is essential for predicting the gene expression of each spot. Our key hypothesis is that the expression levels of certain genes in neighboring regions can strongly indicate the target spot's expression

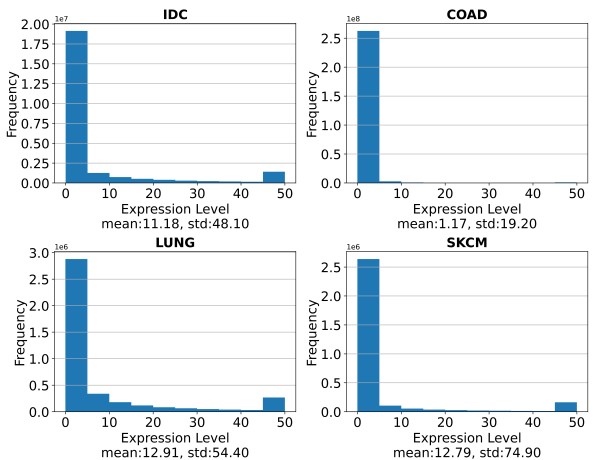

*Figure 2.* Distribution of gene expression level on ST samples, with levels above 50 truncated for clarity.

(Li et al., 2022; Biancalani et al., 2021; Cordell, 2009). However, the standard regression objective cannot model cell-cell interaction as it predicts gene expression in one go. To address this issue, we reformulate the gene expression regression model into a generative model, using samples from a prior distribution as input, which is then iteratively optimized instead of performing a one-step prediction.

Specifically, we apply flow matching (Lipman et al., 2022; Albergo & Vanden-Eijnden, 2022; Jing et al., 2024) as the optimization framework, which aims to learn a denoiser model $f_\theta(\cdot)$:

$$\min_\theta \mathrm{MSE}\left(\boldsymbol{Y}, f_\theta\left(\boldsymbol{Y}_t, \boldsymbol{I}, \boldsymbol{C}, t\right)\right) \tag{2}$$

where $t$ is a time step sampled uniformly from the uniform distribution $[0, 1]$, and $\boldsymbol{Y}_t$ is a linear interpolation between $\boldsymbol{Y}$ and a sample $\boldsymbol{Y}_0$ drawn from a prior distribution $p_0(\cdot)$, i.e., $\boldsymbol{Y}_t = t \cdot \boldsymbol{Y} + (1 - t) \cdot \boldsymbol{Y}_0$. Technically, $f_\theta(\cdot)$ approximates the marginal vector field of the time-dependent conditional probability paths $p_t(\boldsymbol{Y}_t|\boldsymbol{Y})$, enabling generating $\boldsymbol{Y}$ with the noisy sample from $p_0(\cdot)$.

**Prior Distribution** One key advantage of flow matching over diffusion models is its compatibility with different prior distributions. To explore gene expression patterns in ST samples, we analyze certain datasets from HEST-1k (Jaume et al., 2024). Figure 2 shows the distribution of gene expression frequencies across four datasets, revealing two key observations: (1) non-activated genes dominate the dataset, and (2) the data exhibits an overdispersion pattern (variance > mean).

In light of this, we explore zero-inflated negative binomial (ZINB) distribution $\mathcal{Z}(\mu, \phi, \pi)$, defined by the following

probability mass function:

$$p(y \mid \mu, \phi, \pi) =$$
$$\begin{cases} \pi + (1 - \pi) \left(\frac{\Gamma(y+\phi)}{\Gamma(\phi)\, y!}\right) \left(\frac{\phi}{\phi+\mu}\right)^\phi \left(\frac{\mu}{\phi+\mu}\right)^y & \text{if } y = 0, \\ (1 - \pi) \left(\frac{\Gamma(y+\phi)}{\Gamma(\phi)\, y!}\right) \left(\frac{\phi}{\phi+\mu}\right)^\phi \left(\frac{\mu}{\phi+\mu}\right)^y & \text{if } y > 0, \end{cases}$$
$$\tag{3}$$

where $y$ is the count outcome, $\mu$ is the mean of the distribution, $\phi$ denotes the number of failures until stopped, and $\pi$ is the zero-inflation probability. The negative binomial component introduces $\phi$ to explicitly account for variability beyond what is expected under a Poisson distribution, therefore modeling the overdispersion. Besides, the zero-inflation component represents the sparsity with a dropout propability (Virshup et al., 2023; Gayoso et al., 2022; Eraslan et al., 2019). Besides ZINB, Gaussian and zero distributions can also serve as priors, as further explored in Appendix C.

**Training** As shown in Algo.1, during training, we sample a time step $t$ from the uniform distribution and interpolate the ground-truth gene expression $\boldsymbol{Y}$ with the sampled noise $\boldsymbol{Y}_0$ to obtain noisy sample $\boldsymbol{Y}_t$. The denoiser predicts the denoised gene expression with the inputs of image features, coordinates, noisy samples, and time steps. The model is then optimized by minimizing the difference between the prediction and the ground-truth expression.

**Sampling** As shown in Algo.2, we begin with an initial "expression guess" $\boldsymbol{Y}_0$ sampled from the ZINB distribution and iteratively refine it using the trained denoiser. The model interpolates between the noisy input $\boldsymbol{Y}_t$ and the predicted denoised expression $\hat{\boldsymbol{Y}}$ over multiple steps, with a decay coefficient that gradually increases as the time steps increase. This process ultimately converges to the optimal gene expression in the final step.

### 3.3. Denoiser Architecture $f_\theta$

The STFlow's denoiser receives visual features $\boldsymbol{Z}$, coordinates $\boldsymbol{I}$, and gene expression $\boldsymbol{Y}_t$ at time step $t$ as input. The backbone is based on the Transformer architecture (Vaswani, 2017), achieving E(2)-invariance to the coordinates by incorporating frame averaging (FA) within each layer and explicitly encoding spatial dependencies by conducting attention to each spot's local neighbors. The overall architecture is shown in Figure 3(a).

**Local Spatial Context** Cells within the tissues can interact and influence each other's gene expression, thereby forming a spatial context with spot-to-spot dependencies. To efficiently leverage such dependencies, we encode the local spatial context around each spot $i$ and limit the attention to its $k$-nearest neighbors, i.e., $\mathcal{N}(i)$, in the WSI. Long-range context information can be captured through multi-layer

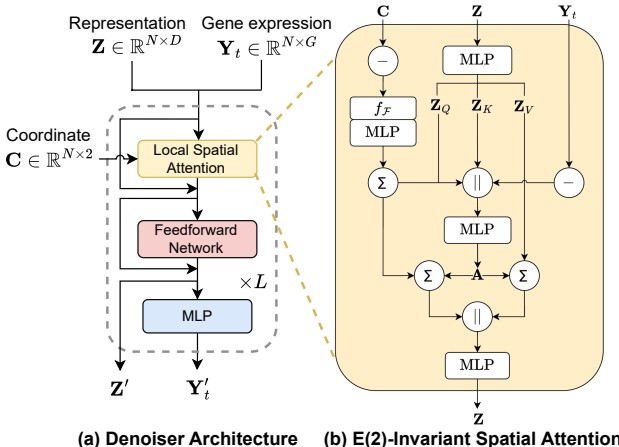

(a) Denoiser Architecture  (b) E(2)-Invariant Spatial Attention

*Figure 3.* The overall architecture of the proposed denoiser (a) and the E(2)-invariant spatial attention scheme (b).

attention within the local neighbors of every spot.

**E(2)-Invariant Spatial Attention** We introduce a spatial attention mechanism that generates spot representations invariant to E(2) operations, i.e., rotation, translation, and reflection, of the coordinates. To achieve this, we adapt frame averaging (FA), an E(2)-invariant transformation for point clouds (Puny et al., 2021), to the attention scheme. The flexibility of FA provides a recipe for encoding the coordinates with minimal modification to the Transformer. Specifically, for $i$-th spot, we first construct the local context with the direction vectors from it to its neighbors:

$$\mathcal{C}_i = \{ \boldsymbol{C}_{i \to j} \mid j \in \mathcal{N}(i) \} \tag{4}$$

where $\boldsymbol{C}_{i \to j} = \boldsymbol{C}_i - \boldsymbol{C}_j$ is the direction vector and represents the orientation between spots. Such a geometric context is then projected into frames extracted by PCA:

$$\mathcal{F}(\mathcal{C}_i) := \{ (\boldsymbol{U}, \hat{\boldsymbol{c}}) \mid \boldsymbol{U} = [\alpha_1 \boldsymbol{u}_1, \alpha_2 \boldsymbol{u}_2], \alpha_{1,2} \in \{-1, 1\} \}, \tag{5}$$

where $\mathcal{F}(\cdot)$ denotes four extracted frames with the two principal components $(\boldsymbol{u}_1, \boldsymbol{u}_2)$ and centroid $\boldsymbol{c}$. We use $\boldsymbol{C}_{i \to j}^{(g)} = (\boldsymbol{C}_{i \to j} - \hat{\boldsymbol{c}})\boldsymbol{U}$ to denote the projected direction vector from $i$-th to $j$-th spot using $g$-th frames. Building on top of them, we embed these spatial spot-spot dependencies with linear layers and achieve invariance by averaging the representations in different frames:

$$\boldsymbol{C}_{i \to j}' = \frac{1}{|\mathcal{F}(\mathcal{C}_i)|} \sum_g \mathrm{MLP}(\boldsymbol{C}_{i \to j}^{(g)}) \tag{6}$$

where $\boldsymbol{C}_{i \to j}' \in \mathbb{R}^d$ is the encoded representation of the spatial relationship between $i$-th spot and its neighbor $j$ at $l$-th layer. With such pairwise encoding, the spatial information

sent from one source spot depends on the target spot, which is compatible with the attention mechanism.

After transforming the image features $\boldsymbol{Z}_i$ into query, key, and value representations: $\boldsymbol{Z}_{Q,i}, \boldsymbol{Z}_{K,i}, \boldsymbol{Z}_{V,i}$, we adopt MLP attention (Brody et al., 2021) to derive the attention weight between spots, which incorporates the spatial information and the gene expression difference between spots within the calculation:

$$\boldsymbol{A}_{ij} =$$

$$\mathrm{Softmax}_i \left( \mathrm{MLP} \left( \boldsymbol{Z}_{Q,i} \,||\, \boldsymbol{Z}_{K,j} \,||\, \boldsymbol{C}_{i \to j}' \,||\, (\boldsymbol{Y}_{t,i} - \boldsymbol{Y}_{t,j}) \right) \right) \tag{7}$$

where $\boldsymbol{A}_{ij}$ denotes the attention score between $i$-th and $j$-th spots, and $\mathrm{Softmax}_i(\cdot)$ is the softmax function operated on the attention scores of spot $i$'s neighbors.

The spatial representation is then aggregated as the context for updating the spot representation, and the gene expression is iteratively updated at each layer, which progressively denoises the gene expression data across different receptive fields:

$$\boldsymbol{Z}_i' = \mathrm{MLP} \left( \sum_{j \in \mathcal{N}(i)} \boldsymbol{A}_{ij} \boldsymbol{Z}_{V,j} \,\Bigg|\Bigg|\, \sum_{j \in \mathcal{N}(i)} \boldsymbol{A}_{ij} \boldsymbol{C}_{i \to j}' \right) + \boldsymbol{Z}_i \tag{8}$$

$$\boldsymbol{Y}_{t,i}' = \mathrm{MLP} \left( \boldsymbol{Z}_i' \right) \tag{9}$$

where $\boldsymbol{Z}_i'$ and $\boldsymbol{Y}_{t,i}'$ represent the updated $i$-th spot's representation and gene expression from the spatial attention module. This process is repeated across each layer, with the gene expression updates from each layer averaged to produce the final gene expression prediction.

### 3.4. Discussion

**Notes on invariance** For the spot-level, Equ.6 demonstrates E(2)-invariance to the coordinates as it encodes and averages the coordinates across different frames, which is guaranteed by the frame averaging framework. Consequently, the spatial attention mechanism (Equ.7 and Equ.9) that relies on the output of Equ.6 is E(2)-invariant. For the pixel level, we apply pathology foundation models, which are pretrained with extensive image augmentations, making the extracted spot features robust to any E(2) transformations.

**Computational Complexity** For spatial attention, FA is efficient due to the low dimensionality of the coordinates (only 2) and the accelerated PCA algorithm, thus we ignore its complexity. The attention calculation involves neighboring spots and linear transformations, resulting in a complexity

*Table 1.* The results of gene expression prediction across two benchmarks. The best result is marked in bold, and the best baseline is underlined. OOM indicates an out-of-memory error.

| | | Spot-based | | | | | Slide-based | | | |
|---|---|---|---|---|---|---|---|---|---|---|
| | | Ciga | UNI | Gigapath | STNet | BLEEP | Gigapath-slide | HisToGene | TRIPLEX | STFlow |
| HEST | IDC | $0.424_{000}$ | $0.520_{001}$ | $0.513_{001}$ | $0.375_{001}$ | $0.533_{001}$ | OOM | $0.355_{003}$ | $0.606_{003}$ | $0.587_{003}$ |
| | PRAD | $0.345_{001}$ | $0.371_{002}$ | $0.384_{000}$ | $0.343_{000}$ | $0.382_{002}$ | $0.385_{000}$ | $0.270_{005}$ | $0.402_{009}$ | $0.421_{002}$ |
| | PAAD | $0.408_{000}$ | $0.432_{002}$ | $0.436_{000}$ | $0.366_{000}$ | $0.459_{000}$ | $0.393_{000}$ | $0.310_{007}$ | $0.492_{000}$ | $0.507_{004}$ |
| | SKCM | $0.493_{001}$ | $0.629_{000}$ | $0.590_{005}$ | $0.381_{002}$ | $0.566_{033}$ | $0.548_{002}$ | $0.321_{008}$ | $0.699_{002}$ | $0.704_{005}$ |
| | COAD | $0.273_{001}$ | $0.285_{002}$ | $0.290_{002}$ | $0.248_{001}$ | $0.303_{007}$ | OOM | $0.070_{006}$ | $0.319_{004}$ | $0.326_{009}$ |
| | READ | $0.051_{002}$ | $0.159_{000}$ | $0.151_{001}$ | $0.097_{000}$ | $0.236_{001}$ | $0.186_{001}$ | $-0.000_{000}$ | $0.195_{006}$ | $0.240_{014}$ |
| | CCRCC | $0.136_{000}$ | $0.178_{002}$ | $0.187_{001}$ | $0.204_{002}$ | $0.298_{003}$ | $0.181_{001}$ | $0.112_{003}$ | $0.289_{005}$ | $0.332_{003}$ |
| | HCC | $0.040_{001}$ | $0.052_{000}$ | $0.051_{000}$ | $0.072_{004}$ | $0.086_{001}$ | $0.025_{001}$ | $0.028_{001}$ | $0.062_{003}$ | $0.124_{004}$ |
| | LUNG | $0.544_{000}$ | $0.559_{001}$ | $0.569_{000}$ | $0.526_{001}$ | $0.588_{004}$ | $0.528_{003}$ | $0.480_{007}$ | $0.601_{002}$ | $0.610_{002}$ |
| | LYMPH | $0.236_{000}$ | $0.263_{000}$ | $0.274_{001}$ | $0.250_{001}$ | $0.234_{004}$ | $0.284_{000}$ | $0.230_{005}$ | $0.292_{002}$ | $0.305_{001}$ |
| Average | | 0.295 | 0.344 | 0.344 | 0.286 | 0.368 | / | 0.237 | 0.395 | **0.415** |
| STImage | Breast | $0.354_{000}$ | $0.412_{000}$ | $0.384_{000}$ | $0.318_{000}$ | $0.131_{001}$ | OOM | $0.172_{019}$ | $0.418_{033}$ | $0.404_{024}$ |
| | Brain | $0.331_{000}$ | $0.344_{001}$ | $0.364_{000}$ | $0.321_{002}$ | $0.322_{004}$ | $0.362_{000}$ | $0.020_{010}$ | $0.292_{034}$ | $0.357_{001}$ |
| | Skin | $0.174_{000}$ | $0.128_{002}$ | $0.235_{001}$ | $0.192_{003}$ | $0.301_{000}$ | $0.174_{000}$ | $0.031_{021}$ | $0.319_{011}$ | $0.310_{011}$ |
| | Mouth | $0.176_{000}$ | $0.177_{001}$ | $0.198_{001}$ | $0.177_{001}$ | $0.191_{000}$ | $0.164_{000}$ | $0.042_{012}$ | $0.107_{013}$ | $0.146_{015}$ |
| | Stomach | $0.169_{000}$ | $0.217_{002}$ | $0.222_{001}$ | $0.213_{004}$ | $0.295_{005}$ | OOM | $0.070_{011}$ | $0.248_{018}$ | $0.305_{041}$ |
| | Prostate | $0.128_{001}$ | $0.163_{001}$ | $0.173_{000}$ | $0.042_{000}$ | $0.167_{028}$ | $0.208_{001}$ | $0.033_{010}$ | $0.148_{021}$ | $0.210_{024}$ |
| | Colon | $0.262_{002}$ | $0.276_{001}$ | $0.307_{000}$ | $0.175_{000}$ | $0.220_{013}$ | $0.268_{002}$ | $0.186_{007}$ | $0.238_{030}$ | $0.323_{015}$ |
| Average | | 0.227 | 0.245 | 0.269 | 0.205 | 0.232 | / | 0.232 | 0.252 | **0.293** |

*Table 2.* Comparison on biomarker prediction.

| | GATA3 | ERBB2 | UBE2C | VWF |
|---|---|---|---|---|
| UNI | $0.838_{006}$ | $0.801_{005}$ | $0.704_{002}$ | $0.572_{007}$ |
| STNet | $0.607_{001}$ | $0.546_{004}$ | $0.662_{000}$ | $0.290_{000}$ |
| BLEEP | $0.806_{004}$ | $0.761_{000}$ | $0.758_{008}$ | $0.535_{096}$ |
| TRIPLEX | $0.853_{008}$ | $0.832_{001}$ | $0.749_{003}$ | $0.612_{017}$ |
| STFlow | $\mathbf{0.860}_{005}$ | $\mathbf{0.844}_{001}$ | $\mathbf{0.772}_{008}$ | $\mathbf{0.666}_{004}$ |
| STFlow w/o FM | $0.842_{005}$ | $0.822_{002}$ | $0.750_{008}$ | $0.570_{001}$ |

of $O(Nkd + Nkd^2)$, where $d$ is the embedding size, and is efficient since $k \ll N$. With flow matching, the computation scales linearly with the number of refinement steps $S$. In practice, this remains efficient as flow matching requires relatively few steps, a key advantage over diffusion models. In our experiments, we set $S$ to 5.

# 4. Experiment

In this section, we evaluate our proposed STFlow for gene expression and biomarker prediction across two datasets, comparing it against 8 baselines. The implementation details can be found in Appendix A, the dataset statistics are shown in Appendix B, and more studies are presented in Appendix C.

## 4.1. Gene Expression Prediction

**Datasets** We employ two public collections of spatial transcriptomics data paired with H&E-stained WSIs, HEST-1k (Jaume et al., 2024) and STImage-1K4M (Chen et al., 2024a). Specifically, HEST-1k includes ten benchmarks[2] and applies a patient-stratified split to prevent data leakage, resulting in a $k$-fold cross-validation setup. For STImage-1K4M, we select the cancer samples for each organ and randomly split the dataset into train/val/test sets (8:1:1). Performance is evaluated using the Pearson correlation between the predicted and measured gene expressions for the top 50 highly variable genes after log1p normalization. We conduct the experiments using three different random seeds and report the mean and standard deviation for each dataset.

**Baselines** There are two categories of methods:

- Spot-based approaches, including Ciga (Ciga et al., 2022), UNI (Chen et al., 2024b), Gigapath (Xu et al., 2024), STNet (He et al., 2020), and BLEEP (Xie et al., 2023), predict the gene expression solely based on the spot image. BLEEP applies UNI as the image encoder. For pathology foundation models, we use a Random Forest model as the regression head, following the setup of HEST-1k.

- Slide-based approaches, including Gigapath-slide which is the slide encoder of Gigapath, HisToGene (Pang et al., 2021), and TRIPLEX (Chung et al., 2024), incorporate the whole-slide information by aggregating the local or global context around each spot. The coordinates are embedded

---

[2]Note that the COAD dataset was updated after the paper's release, leading to a significant difference in the performance reported in the HEST-1k manuscript.

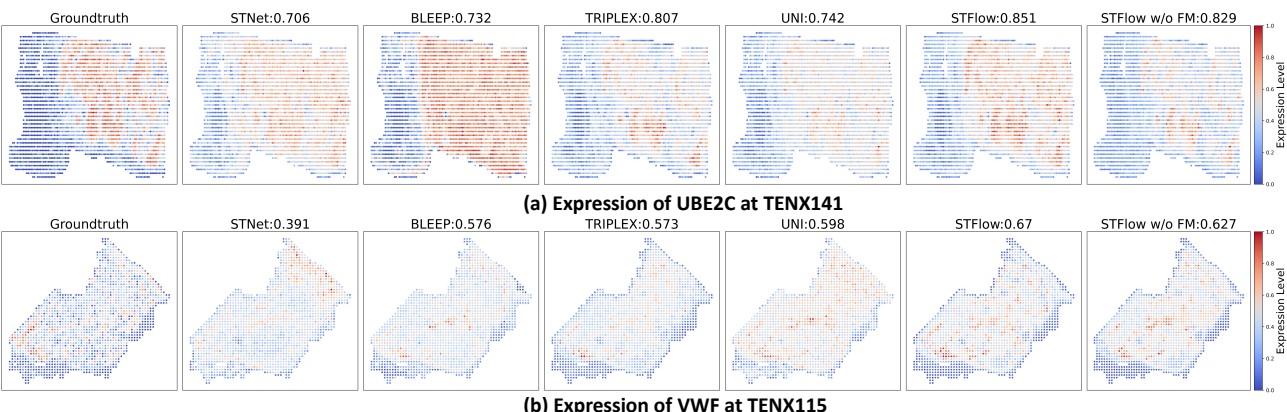

*Figure 4.* STFlow for Biomarker Discovery in Breast Samples: **(a)** TENX141 and **(b)** TENX115. The top row of subfigures shows gene UB2EC, while the bottom row shows gene VWF. The Pearson correlation between ground truth and predictions is provided in each subfigure's title. STFlow w/o FM refers to the model where the iterative refinement is removed and one-step prediction is performed.

using a linear layer or a convolution layer, serving as position encoding. STFlow and TRIPLEX apply UNI as the image encoder.

**Results** As shown in Table 1, even with a simple linear head, the pathology foundation models demonstrate a significant correlation over most datasets. Besides, building on these foundation models, our proposed STFlow can reach better performance and achieve 18% improvement on average, highlighting its compatibility with the pathology foundation models and demonstrating the effectiveness of leveraging spatial context and gene interaction.

Additionally, some approaches fail to achieve strong gene expression prediction performance. For example, STNet (PCC = 0.286) and HisToGene (PCC = 0.237) both underperform compared to UNI (PCC = 0.344). We attribute this to the patient-level data split, which introduces a more challenging scenario than previous splits, making it difficult for these methods to capture the meaningful semantics of the spot images. This observation is consistent with the findings in (Chung et al., 2024). This also highlights the advantage of using a foundation model and the need for an effective spatial encoder. Furthermore, Gigapath-slide, which aggregates whole-slide information, does not outperform Gigapath in most tasks. This may be because the slide encoder's pretrained objective is tailored for slide-level tasks rather than spot-level tasks.

### 4.2. Biomarker Prediction

One of the important applications of spatial gene expression prediction is to understand disease progression in relation to tissue morphology. In this section, we evaluate model performance in predicting four biomarker genes: GATA3, ERBB2, UBE2C, and VWF (Mehra et al., 2005; Revillion et al., 1998; Ma et al., 2023; Dachy et al., 2024; Takaya

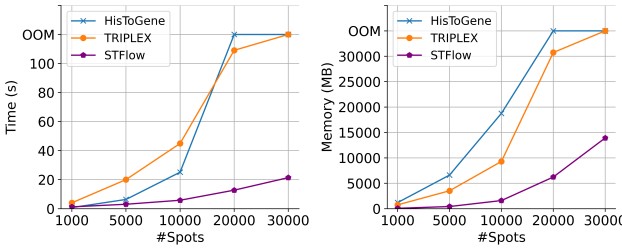

*Figure 5.* Efficiency comparison of slide-based models.

et al., 2018). We utilize two datasets from HEST and report the average correlation for each gene across different cross-validation folds. Specifically, the IDC dataset is used for GATA3 and ERBB2, while LUNG is used for UBE2C, and SKCM for VWF. All results are averaged over three random seeds and the default visual extractor is UNI. In Table 2, STFlow achieves the highest correlation across all biomarkers, demonstrating its potential for clinical applications.

Additionally, we visualize the expression levels of UBE2C and VWF in two samples (Figure 4). The results further highlight the strong correlation between STFlow's predictions and ground-truth gene expression. Taking UBE2C in TENX141 as an example, flow matching effectively optimizes high-expression regions (red areas) more accurately than the model without flow matching.

### 4.3. Further Analysis on STFlow

**Efficiency Comparison** To evaluate the efficiency of STFlow, we compare the total inference time and GPU memory usage across slide-based models with varying numbers of spots, as shown in Figure 5. Inference time is reported as the total duration for 100 repeated runs. For a fair comparison, we exclude the time required for visual feature extraction for all models. Notably, compared to

*Table 3.* E(2)-invariant architecture and slide-based models comparison using different pathology foundation models. The complete results can be found in Appendix C.

| | E(2)-based Arch. | | SOTA Baselines | | STFlow |
| --- | --- | --- | --- | --- | --- |
| | EGNN | E2CNN | BLEEP | TRIPLEX | |
| Ciga | 0.314 | 0.334 | 0.323 | 0.342 | **0.361** |
| UNI | 0.400 | 0.363 | 0.368 | 0.395 | **0.415** |
| Gigapath | 0.391 | 0.375 | 0.369 | 0.399 | **0.406** |

*Table 4.* Ablation study on STFlow. The complete results can be found in Appendix C.

| | STFlow w/o FM | STFlow w/o FA | STFlow |
| --- | --- | --- | --- |
| Ciga | 0.351 | 0.348 | **0.361** |
| UNI | 0.405 | 0.406 | **0.415** |
| Gigapath | 0.397 | 0.397 | **0.406** |

other architectures that rely on complicated attention mechanisms, our proposed architecture achieves higher efficiency by leveraging spatial attention among local neighbors.

**E(2)-Invariant Architecture Comparison** To verify the effectiveness of our proposed E(2)-invariant denoiser, we replace our proposed architecture with two representative E($n$)-invariant architectures individually:

- EGNN (Satorras et al., 2021) is a representative E($n$) graph neural network that leverages invariant geometric feature distance between coordinates to ensure representation invariance. The model conducts representation aggregation among the $k$-nearest neighbors for each spot. For a fair comparison, EGNN also receives the input of extracted image features as spot features.

- E2CNN (Weiler & Cesa, 2019) is a representative framework for E($n$) convolutional neural networks that utilizes irreducible representations. We use the extracted features as input channels and construct a tensor of neighboring spots centered around the target spots. This tensor is then fed into a ResNet model built using E2CNN.

More description and implementation regarding these two methods can be found at Appendix A. The comparison results are presented in Table 3, where we only show the average results across the benchmark datasets of HEST-1k due to the space limit. The results show that STFlow's performance decreases to varying degrees when using EGNN or E2CNN in all cases. We attribute this to the fact that the geometric features used in these models are either simple, as in the case of distances in EGNN, or extracted through constrained functions, such as group steerable kernels in E2CNN. In contrast, FA-based transformation directly leverages the direction vectors, allowing the model to automatically learn relevant geometric features in the latent space.

**Effect on Pathology Foundation Model** As presented in Table 3, we further compare STFlow with the SOTA methods using different pathology foundation models. The results demonstrate a consistent improvement of STFlow over the pathology foundation models, highlighting its compatibility. Moreover, STFlow consistently outperforms all

baselines, showcasing the effectiveness of modeling cell-cell interactions with flow matching and the expressiveness of our proposed architecture.

**Ablation Studies** We conduct an ablation study to assess the effectiveness of the flow matching framework and frame averaging-based modules. Specifically, we evaluate two settings: disabling the flow matching learning framework (w/o FM) and removing frame averaging-related transformations (w/o FA). Notably, without FA-based modules, the architecture reverts to a standard attention scheme without utilizing spatial information. The results presented in Table 4 indicate that removing either core module leads to a decline in STFlow's performance.

Furthermore, we evaluate the impact of removing flow matching on the biomarker prediction task, as shown in Table 2. The results demonstrate the importance of flow matching, as its absence leads to performance degradation. As illustrated in Figure 4, the iterative refinement process enhances the alignment between predicted and ground-truth expression levels, resulting in improved correlation.

**Effect on Number of Refinement Steps** We vary the number of refinement steps ($S$) in the range $1, 2, 5, 10, 16$ and report the corresponding performance of STFlow in Table 5. It should be noted that $S = 1$ corresponds to a model without iterative refinement, performing a single-step prediction. The results demonstrate a clear performance improvement when moving from one-step prediction ($S = 1$) to iterative refinement ($S = 2$). In certain datasets, performance continues to improve up to $S = 5$ (e.g., PAAD: $0.488 \rightarrow 0.507$; IDC: $0.580 \rightarrow 0.587$). However, beyond $S = 5$, the gains plateau or slightly decline. This trend is consistent with prior studies such as AlphaFlow (Jing et al., 2024) and RNAFlow (Nori & Jin, 2024), which also adopt $S = 5$ as the default setting.

During the inference process of flow matching, the denoiser's output is interpolated with the noisy input at each step using a decay coefficient, allowing for gradual convergence toward an accurate prediction. Figure 6 presents two breast cancer examples illustrating STFlow's refinement process for gene expression prediction. The process begins with a randomly initialized prediction $\boldsymbol{Y}_0$, which is progressively refined through iterative denoising steps. As refinement

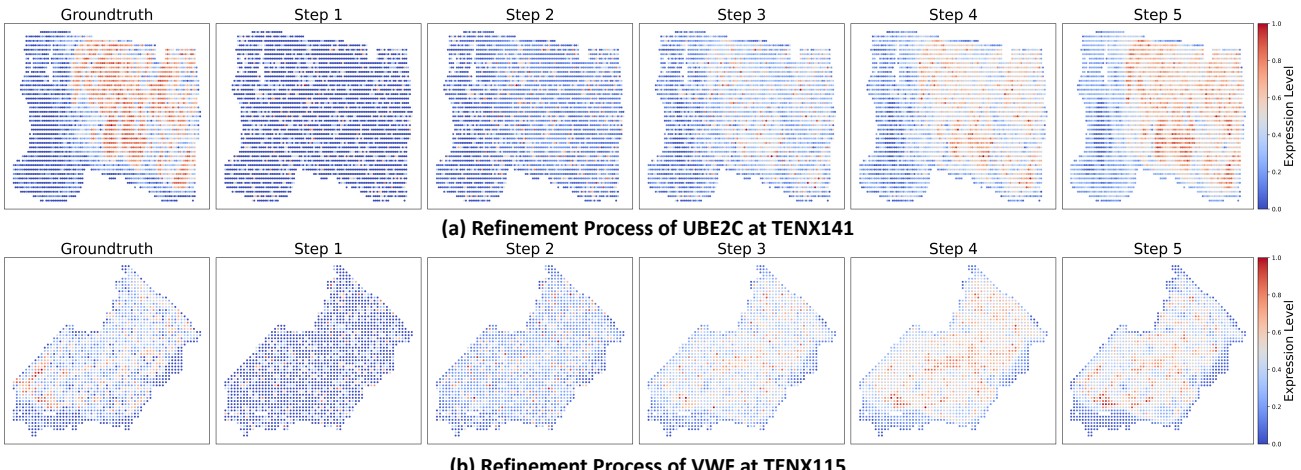

*Figure 6.* Two breast cancer examples of the refinement process performed by STFlow: **(a)** TENX141 and **(b)** TENX115. The top row of subfigures shows gene UB2EC, while the bottom row shows gene VWF.

*Table 5.* The performance of STFlow with different numbers of refinement steps, where $S = 1$ denotes one-step prediction.

|        | $S = 1$ | $S = 2$ | $S = 5$ | $S = 10$ | $S = 16$ |
|--------|---------|---------|---------|----------|----------|
| IDC    | $0.580_{.005}$ | $0.585_{.002}$ | $0.587_{.003}$ | $0.585_{.001}$ | $0.585_{.001}$ |
| PRAD   | $0.420_{.003}$ | $0.416_{.003}$ | $0.421_{.002}$ | $0.414_{.003}$ | $0.415_{.004}$ |
| PAAD   | $0.488_{.001}$ | $0.498_{.001}$ | $0.507_{.004}$ | $0.499_{.001}$ | $0.498_{.001}$ |
| SKCM   | $0.705_{.002}$ | $0.707_{.005}$ | $0.704_{.005}$ | $0.703_{.005}$ | $0.703_{.005}$ |
| COAD   | $0.315_{.008}$ | $0.343_{.004}$ | $0.326_{.009}$ | $0.320_{.003}$ | $0.321_{.004}$ |
| READ   | $0.232_{.009}$ | $0.239_{.002}$ | $0.240_{.014}$ | $0.239_{.003}$ | $0.239_{.004}$ |
| CCRCC  | $0.322_{.001}$ | $0.340_{.002}$ | $0.332_{.003}$ | $0.330_{.002}$ | $0.319_{.003}$ |
| HCC    | $0.115_{.008}$ | $0.119_{.002}$ | $0.124_{.004}$ | $0.117_{.003}$ | $0.118_{.002}$ |
| LUNG   | $0.604_{.002}$ | $0.612_{.002}$ | $0.610_{.002}$ | $0.611_{.002}$ | $0.611_{.001}$ |
| LYMPH  | $0.278_{.002}$ | $0.310_{.001}$ | $0.305_{.001}$ | $0.306_{.001}$ | $0.305_{.001}$ |
| Average | 0.405 | 0.417 | 0.415 | 0.412 | 0.411 |

proceeds, the predicted expression patterns increasingly resemble the ground truth, demonstrating STFlow 's ability to recover biologically meaningful spatial gene expression through interpolation guided by the decay coefficient.

## 5. Conclusion

In this paper, we study the problem of gene expression prediction from histology images. Despite the promising results achieved by the previous methods, we argue that cell interaction, which is a key factor regulating gene expression, has been overlooked. Motivated by this, we propose STFlow, a flow matching framework incorporating gene-gene dependency with an iterative refinement paradigm. We explore the zero-inflated negative binomial distribution as the prior distribution, which utilizes the inductive bias of the gene expression data. Specifically, the denoiser architecture is a frame-averaging Transformer that integrates spatial context and gene interactions within the attention mechanism. Our experimental results across 2 benchmarks (HEST-1k (Jaume et al., 2024) and STImage-1K4M (Chen et al., 2024a)), including 17 datasets and 4 biomarker genes, show that STFlow consistently outperforms the SOTA baseline methods.

**Limitation** Our learning framework does not include the estimation of the hyperparameters for the ZINB distribution; instead, we use a grid search to identify the optimal hyperparameter combination. A potential improvement would be to initially employ the empirical distribution or a distribution estimation model, such as a Variational Autoencoder (VAE), to estimate the hyperparameters based on the training set, which will be explored for future work.

## Acknowledgment

We thank Yangtian Zhang, Weikang Qiu, and anonymous reviewers for their valuable feedback on the manuscript. This work was supported by the BroadIgnite Award, the Eric and Wendy Schmidt Center at the Broad Institute of MIT and Harvard, NSF IIS Div Of Information & Intelligent Systems 2403317, Amazon research, Yale AI Engineering Research Seed Grants, Yale Office of the Provost, and Yale Wu Tsai Institute.

## Impact Statement

This paper presents work whose goal is to advance the field of spatial transcriptomics prediction on histology images. All the datasets used in this study are publicly available. There are some potential societal consequences of our work, none of which we feel must be specifically highlighted here.

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

# A. Implementation

**Running environment**    The experiments are conducted on a single Linux server with The AMD EPYC 7763 64-Core Processor, 1024G RAM, and 8 RTX A6000-48GB. Our method is implemented on PyTorch 2.3.0 and Python 3.10.14.

**Training details**    For all the models, we fix the optimizer as Adam (Kingma & Ba, 2014) and MSE loss as the loss function. The gradient norm is clipped to 1.0 in each training step to ensure learning stability. The learning rate is tuned within {1e-3, 5e-4, 1e-4} and is set to 5e-4 by default, as it generally yields the best performance. The performance of HEST-1k is evaluated using a cross-validation setup and reported based on three different random seeds. For STImage-1K4M, performance on the test set is reported using the model that achieves the best validation set performance, also evaluated across three random seeds. Besides, all the weights of pathology foundation models are frozen.

For each model, we search the hyperparameters in the following ranges: the dropout rate in {0, 0.2, 0.5}, the number of nearest neighbors for the slide-based methods in {4, 8, 25}, and the number of attention heads in {1, 2, 4, 8}. All models are trained for 100 epochs, with early stopping applied if no performance improvement is observed for 20 epochs. The implementation and hyperparameters used in each method are shown below:

- STFlow: The number of layers, attention heads, and neighbors are 4, 4, and 8, respectively. Besides, dropout and hidden sizes are set at 0.2 and 128. The number of sampling steps for flow matching is set to 5. For the ZINB distribution, zero-inflation probability is fixed as 0.5, the mean is searched {0.1,0.2,0.4}, and the number of failures is searched in {1,2,4}. For efficient training, each sample is a randomly selected continuous region from the WSI, with its size determined by a proportion sampled from a uniform distribution ranging from 0 to 1. In each training step, we will sample a region from WSI.

- Ciga[3], UNI[4], and Gigapath[5]: We download the pretrained weight from the official repository and normalize the input images using the ImageNet mean and standard deviation. The Random Forest model with 70 trees serves as the linear head. Additionally, Gigapath offers three different pretrained versions; we selected the one with the largest hidden size, i.e., "gigapath_slide_enc12l1536d". For the Gigapath-slide, all the spot images of a WSI are input for global attention.

- STNet[6]: Following the official implementation, we use a pretrained DenseNet121 as the image encoder and an MLP as the linear head. The input spot images are randomly augmented with horizontal flips and rotations and are then normalized using the ImageNet mean and standard deviation. The batch size (the number of spot images in each training step) is 128.

- BLEEP[7]: This method trains an image encoder and a gene expression encoder using contrastive loss. For a given spot image, it retrieves the gene expressions of similar spots from a reference set, using the average expression of these spots as the prediction. We use a pretrained pathology foundation as the image encoder and MLPs as linear heads to project the extracted visual features and gene expressions. The temperature for the contrastive loss is set to 1, and the number of retrieved spots is 50. For a fair comparison, we directly use the training WSI as the reference set, as there are no additional splits in HEST-1k and STImage-1K4M. The batch size is set as 128.

- HisToGene[8]: This model includes a ViT for encoding spot images within the WSI. The number of layers, the number of attention heads, the dropout rate, and the hidden size are set as 4, 16, 0.1, and 128. The coordinates are embedded with a linear layer. The input spot images are randomly augmented with horizontal flips and rotations and are then normalized using the ImageNet mean and standard deviation. For efficient training, we sample a continuous region from WSI in each training step, similar to STFlow.

- TRIPLEX[9]: This model comprises a target encoder for the target spot, a local encoder for the neighboring spots, a global encoder for WSI, and a fusion encoder for combining all these representations. In line with the official implementation, the spot images are first embedded using a pathology foundation model before being fed into the model. Each encoder is configured with 2 layers, 8 attention heads, and a dropout rate of 0.1. The local encoder considers 25 neighboring spots.

---

[3]https://github.com/ozanciga/self-supervised-histopathology
[4]https://huggingface.co/MahmoodLab/UNI
[5]https://huggingface.co/prov-gigapath/prov-gigapath
[6]https://github.com/bryanhe/ST-Net/tree/master
[7]https://github.com/bowang-lab/BLEEP/tree/main
[8]https://github.com/maxpmx/HisToGene
[9]https://github.com/NEXGEM/TRIPLEX/tree/main

The coordinates are embedded using a proposed atypical position encoding generator based on a convolutional network. A continuous region from WSI is sampled for each training step, using the same strategy as STFlow.

The number of model parameters is presented in Table 6, highlighting that our model has the lowest parameter count among the slide-level baselines. Specifically, HistToGene utilizes dedicated image encoders, significantly increasing their parameter count. TRIPLEX employs a three-branch attention mechanism, which is substantially more complex and heavier compared to our proposed spatial attention model.

*Table 6.* Number of parameters comparison.

|  | STNet | BLEEP | HistToGene | TRIPLEX | STFlow |
|---|---|---|---|---|---|
| #Params (M) | 0.051 | 0.670 | 149.046 | 13.767 | 1.147 |

We also provide the implementation of the E(2)-invariant encoder baselines:

- EGNN[10]: Similar to a standard GNN, EGNN propagates representations from neighboring spots to the target spots and uses MLPs for transformation, incorporating the distances between them in the calculations. The number of layers and neighbors is set to 4 and 8, respectively, with a hidden size of 128 and a dropout rate of 0.2. For a fair comparison, EGNN leverages the visual features extracted by the pathology foundation model and is integrated with the flow matching.

- E2CNN[11]: E2CNN is an $E(n)$ convolution framework that implements various equivariant operations, such as convolution layers, batchnorm, and pooling layers. Here, we use the 10-layer ResNet from the official codebase as the backbone. To construct the input batch, each spot and its surrounding neighbors are arranged into a $5 \times 5$ grid with the target spot at the center. The visual features extracted by the foundation models are then stacked as channels, resulting in a tensor of dimensions $d \times 5 \times 5$.

## B. Dataset

Table 7 lists the statistics of HEST-Bench. Further details about these datasets can be found in (Jaume et al., 2024). Note that the COAD dataset differs from the version in (Jaume et al., 2024), as it was updated two months after the paper's release.

*Table 7.* Dataset statistics of HEST-Bench.

|  | IDC | PRAD | PAAD | SKCM | COAD | READ | CCRCC | HCC | LUNG | LYMPH |
|---|---|---|---|---|---|---|---|---|---|---|
| Organ | Breast | Prostate | Pancreas | Skin | Colon | Rectum | Kidney | Liver | Lung | Axillary Lymph Nodes |
| Technology | Xenium | Visium | Xenium | Xenium | Visium | Visium | Visium | Visium | Xenium | Visium |
| #Patients | 4 | 2 | 3 | 2 | 3 | 2 | 24 | 2 | 2 | 4 |
| #Samples | 4 | 23 | 3 | 2 | 6 | 4 | 24 | 2 | 2 | 4 |
| #Splits | 4 | 2 | 3 | 2 | 2 | 2 | 6 | 2 | 2 | 4 |
| Avg. spots | 4925 | 2454 | 2780 | 1741 | 5079 | 1909 | 2792 | 1941 | 1944 | 4990 |

We additionally construct 7 benchmark datasets based on STImage-1K4M (Chen et al., 2024a). Specifically, we select cancer samples for each organ and randomly split the train/validation/test sets (8:1:1) at the slide level, which is different from the cross-validation setting used in HEST-Bench. Note that we only include organs where the models achieve significant correlations (at least $> 0.1$). The model predicts the top 50 highly variable genes, with Pearson correlation used as the evaluation metric. The statistics are shown in Table 8.

*Table 8.* Dataset statistics of STImage-Bench.

|  | Breast | Brain | Skin | Mouth | Prostate | Stomach | Colon |
|---|---|---|---|---|---|---|---|
| Technology | Visium | Visium | Visium | Visium | Visium | Visium | Visium |
| #Samples | 81 | 22 | 4 | 16 | 7 | 12 | 4 |
| Avg. spots | 1762 | 2156 | 2232 | 2730 | 3326 | 3117 | 4024 |

[10]https://github.com/vgsatorras/egnn
[11]https://github.com/QUVA-Lab/e2cnn/tree/master

## C. Additional Experiments

**Ablation study** In this experiment, we perform an ablation study to evaluate the impact of STFlow's core modules. Specifically, we individually disable the flow matching learning framework (w/o FM) and the frame averaging-related transformations (w/o FA). As shown in the Table 9, we can observe that removing any of the core modules leads to performance degradation to varying degrees, and this trend remains consistent across different pathology foundation models.

*Table 9.* Ablation study.

| | | IDC | PRAD | PAAD | SKCM | COAD | READ | CCRCC | HCC | LUNG | LYMPH | Avg. |
|---|---|---|---|---|---|---|---|---|---|---|---|---|
| Ciga | STFlow | $0.460_{028}$ | $0.375_{000}$ | $0.440_{005}$ | $0.602_{002}$ | $0.334_{006}$ | $0.137_{004}$ | $0.250_{054}$ | $0.094_{001}$ | $0.583_{000}$ | $0.308_{000}$ | 0.361 |
| | w/o FM | $0.436_{001}$ | $0.380_{003}$ | $0.420_{004}$ | $0.595_{004}$ | $0.336_{008}$ | $0.124_{002}$ | $0.245_{043}$ | $0.096_{000}$ | $0.585_{001}$ | $0.295_{001}$ | 0.351 |
| | w/o FA | $0.446_{020}$ | $0.371_{003}$ | $0.434_{007}$ | $0.585_{006}$ | $0.330_{007}$ | $0.120_{017}$ | $0.235_{002}$ | $0.090_{003}$ | $0.580_{003}$ | $0.295_{002}$ | 0.348 |
| Gigapath | STFlow | $0.565_{000}$ | $0.415_{001}$ | $0.510_{000}$ | $0.652_{001}$ | $0.315_{007}$ | $0.257_{006}$ | $0.326_{003}$ | $0.117_{001}$ | $0.602_{000}$ | $0.305_{001}$ | 0.406 |
| | w/o FM | $0.560_{005}$ | $0.409_{003}$ | $0.504_{003}$ | $0.656_{005}$ | $0.296_{008}$ | $0.225_{006}$ | $0.325_{004}$ | $0.120_{001}$ | $0.595_{002}$ | $0.285_{002}$ | 0.397 |
| | w/o FA | $0.560_{000}$ | $0.418_{000}$ | $0.504_{001}$ | $0.615_{000}$ | $0.326_{005}$ | $0.250_{008}$ | $0.302_{003}$ | $0.111_{001}$ | $0.593_{003}$ | $0.291_{003}$ | 0.397 |
| UNI | STFlow | $0.587_{003}$ | $0.421_{002}$ | $0.507_{004}$ | $0.704_{005}$ | $0.326_{009}$ | $0.240_{014}$ | $0.332_{003}$ | $0.124_{004}$ | $0.610_{002}$ | $0.305_{001}$ | 0.415 |
| | w/o FM | $0.580_{005}$ | $0.420_{003}$ | $0.488_{001}$ | $0.705_{002}$ | $0.315_{008}$ | $0.232_{009}$ | $0.322_{001}$ | $0.115_{008}$ | $0.604_{002}$ | $0.278_{002}$ | 0.405 |
| | w/o FA | $0.580_{005}$ | $0.421_{001}$ | $0.501_{003}$ | $0.673_{006}$ | $0.323_{002}$ | $0.245_{015}$ | $0.305_{006}$ | $0.113_{004}$ | $0.600_{001}$ | $0.299_{005}$ | 0.406 |

**Prior distribution comparison** We conduct an experiment to investigate the influence of different prior distributions used in STFlow. Specifically, we replace the ZINB distribution with two alternatives: zero distribution, where all samples are zero, and standard Gaussian distribution. The results are summarized in Table 10, from which we can observe that the ZINB distribution consistently achieves the best performance across all cases. This demonstrates that it is better suited to represent gene expression data, which is often sparse and overdispersed. Note that comparing the average Pearson correlation across datasets might underestimate the contribution of ZINB due to significant scale differences between datasets.

*Table 10.* Prior distribution comparison on STFlow.

| | Gaussian | | | Zero | | | ZINB | | |
|---|---|---|---|---|---|---|---|---|---|
| | Ciga | UNI | Gigapath | Ciga | UNI | Gigapath | Ciga | UNI | Gigapath |
| IDC | $0.451_{000}$ | $0.580_{001}$ | $0.559_{001}$ | $0.450_{006}$ | $0.585_{003}$ | $0.567_{000}$ | $0.460_{028}$ | $0.587_{003}$ | $0.565_{000}$ |
| PRAD | $0.370_{000}$ | $0.410_{004}$ | $0.410_{003}$ | $0.370_{005}$ | $0.410_{002}$ | $0.414_{003}$ | $0.375_{000}$ | $0.421_{002}$ | $0.415_{001}$ |
| PAAD | $0.430_{000}$ | $0.500_{001}$ | $0.507_{003}$ | $0.424_{009}$ | $0.496_{001}$ | $0.506_{000}$ | $0.440_{005}$ | $0.507_{004}$ | $0.510_{000}$ |
| SKCM | $0.600_{002}$ | $0.698_{003}$ | $0.643_{005}$ | $0.600_{001}$ | $0.698_{002}$ | $0.654_{001}$ | $0.602_{002}$ | $0.704_{005}$ | $0.652_{001}$ |
| COAD | $0.337_{007}$ | $0.317_{006}$ | $0.320_{006}$ | $0.332_{002}$ | $0.319_{005}$ | $0.316_{004}$ | $0.334_{006}$ | $0.326_{009}$ | $0.315_{007}$ |
| READ | $0.043_{003}$ | $0.226_{013}$ | $0.253_{000}$ | $0.130_{000}$ | $0.230_{006}$ | $0.265_{001}$ | $0.137_{004}$ | $0.240_{014}$ | $0.257_{006}$ |
| CCRCC | $0.259_{004}$ | $0.326_{003}$ | $0.320_{002}$ | $0.233_{001}$ | $0.319_{002}$ | $0.324_{002}$ | $0.250_{054}$ | $0.332_{003}$ | $0.326_{003}$ |
| HCC | $0.103_{000}$ | $0.115_{001}$ | $0.115_{000}$ | $0.095_{000}$ | $0.115_{002}$ | $0.114_{004}$ | $0.094_{001}$ | $0.124_{004}$ | $0.117_{001}$ |
| LUNG | $0.575_{000}$ | $0.605_{001}$ | $0.594_{003}$ | $0.580_{001}$ | $0.604_{002}$ | $0.596_{003}$ | $0.583_{000}$ | $0.610_{002}$ | $0.602_{000}$ |
| LYMPH | $0.300_{002}$ | $0.302_{000}$ | $0.302_{001}$ | $0.301_{005}$ | $0.300_{000}$ | $0.301_{002}$ | $0.308_{000}$ | $0.305_{001}$ | $0.305_{001}$ |
| Average | 0.347 | 0.407 | 0.402 | 0.351 | 0.407 | 0.405 | 0.361 | 0.415 | 0.406 |

**Comparison of Architecture and Pathology Foundation Model** Table 11 presents the complete results of E(2)-invariant architectures, i.e., EGNN and E2CNN, and Table 12 shows the results of state-of-the-art baselines, i.e., BLEEP and TRIPLEX, on each dataset. The performance of each model using different pathology foundation models is presented.

*Table 11.* E(2)-invariant architecture comparison.

| | EGNN | | | E2CNN | | | STFlow | | |
|---|---|---|---|---|---|---|---|---|---|
| | Ciga | UNI | Gigapath | Ciga | UNI | Gigapath | Ciga | UNI | Gigapath |
| IDC | $0.450_{041}$ | $0.575_{006}$ | $0.565_{007}$ | $0.451_{002}$ | $0.562_{002}$ | $0.547_{006}$ | $0.460_{028}$ | $0.587_{003}$ | $0.565_{000}$ |
| PRAD | $0.200_{000}$ | $0.409_{002}$ | $0.410_{007}$ | $0.300_{002}$ | $0.240_{001}$ | $0.372_{003}$ | $0.375_{000}$ | $0.421_{002}$ | $0.415_{001}$ |
| PAAD | $0.420_{001}$ | $0.492_{007}$ | $0.505_{004}$ | $0.442_{006}$ | $0.458_{005}$ | $0.470_{002}$ | $0.440_{005}$ | $0.507_{004}$ | $0.510_{000}$ |
| SKCM | $0.571_{008}$ | $0.661_{008}$ | $0.600_{009}$ | $0.572_{003}$ | $0.674_{003}$ | $0.621_{002}$ | $0.602_{002}$ | $0.704_{005}$ | $0.652_{001}$ |
| COAD | $0.343_{002}$ | $0.321_{001}$ | $0.324_{001}$ | $0.335_{001}$ | $0.329_{007}$ | $0.307_{004}$ | $0.334_{006}$ | $0.326_{009}$ | $0.315_{007}$ |
| READ | $0.118_{004}$ | $0.238_{001}$ | $0.230_{006}$ | $0.119_{001}$ | $0.221_{010}$ | $0.223_{005}$ | $0.137_{004}$ | $0.240_{014}$ | $0.257_{006}$ |
| CCRCC | $0.091_{005}$ | $0.317_{003}$ | $0.296_{003}$ | $0.268_{001}$ | $0.300_{002}$ | $0.292_{008}$ | $0.250_{054}$ | $0.332_{003}$ | $0.326_{003}$ |
| HCC | $0.095_{006}$ | $0.100_{006}$ | $0.104_{004}$ | $0.060_{002}$ | $0.090_{009}$ | $0.103_{004}$ | $0.094_{001}$ | $0.124_{004}$ | $0.117_{001}$ |
| LUNG | $0.555_{005}$ | $0.592_{004}$ | $0.584_{008}$ | $0.502_{001}$ | $0.499_{002}$ | $0.550_{002}$ | $0.583_{000}$ | $0.610_{002}$ | $0.602_{000}$ |
| LYMPH | $0.302_{001}$ | $0.290_{001}$ | $0.293_{006}$ | $0.292_{001}$ | $0.260_{005}$ | $0.274_{001}$ | $0.308_{000}$ | $0.305_{001}$ | $0.305_{001}$ |
| Average | 0.314 | 0.400 | 0.391 | 0.334 | 0.363 | 0.375 | 0.361 | 0.415 | 0.406 |

*Table 12.* The comparison between models using different pathology foundation models.

|  | BLEEP | | | TRIPLEX | | | STFlow | | |
|---|---|---|---|---|---|---|---|---|---|
|  | Ciga | UNI | Gigapath | Ciga | UNI | Gigapath | Ciga | UNI | Gigapath |
| IDC | $0.432_{001}$ | $0.533_{001}$ | $0.519_{000}$ | $0.493_{000}$ | $0.606_{003}$ | $0.591_{002}$ | $0.460_{028}$ | $0.587_{003}$ | $0.565_{000}$ |
| PRAD | $0.337_{000}$ | $0.382_{002}$ | $0.386_{001}$ | $0.353_{000}$ | $0.402_{009}$ | $0.405_{000}$ | $0.375_{000}$ | $0.421_{002}$ | $0.415_{001}$ |
| PAAD | $0.411_{004}$ | $0.459_{000}$ | $0.473_{001}$ | $0.430_{004}$ | $0.492_{000}$ | $0.505_{000}$ | $0.440_{005}$ | $0.507_{004}$ | $0.510_{000}$ |
| SKCM | $0.542_{008}$ | $0.566_{033}$ | $0.571_{002}$ | $0.580_{006}$ | $0.699_{002}$ | $0.670_{002}$ | $0.602_{002}$ | $0.704_{005}$ | $0.652_{001}$ |
| COAD | $0.295_{013}$ | $0.303_{007}$ | $0.301_{000}$ | $0.308_{001}$ | $0.319_{004}$ | $0.315_{002}$ | $0.334_{006}$ | $0.326_{009}$ | $0.315_{007}$ |
| READ | $0.133_{001}$ | $0.236_{001}$ | $0.251_{009}$ | $0.125_{000}$ | $0.195_{006}$ | $0.212_{005}$ | $0.137_{004}$ | $0.240_{014}$ | $0.257_{006}$ |
| CCRCC | $0.224_{004}$ | $0.298_{003}$ | $0.270_{015}$ | $0.240_{001}$ | $0.289_{005}$ | $0.291_{002}$ | $0.250_{054}$ | $0.332_{003}$ | $0.326_{003}$ |
| HCC | $0.071_{002}$ | $0.086_{001}$ | $0.094_{001}$ | $0.046_{000}$ | $0.062_{002}$ | $0.101_{003}$ | $0.094_{001}$ | $0.124_{004}$ | $0.117_{001}$ |
| LUNG | $0.537_{005}$ | $0.588_{004}$ | $0.578_{002}$ | $0.554_{000}$ | $0.601_{002}$ | $0.606_{000}$ | $0.583_{000}$ | $0.610_{002}$ | $0.602_{000}$ |
| LYMPH | $0.252_{008}$ | $0.234_{004}$ | $0.248_{005}$ | $0.297_{003}$ | $0.292_{002}$ | $0.300_{001}$ | $0.308_{000}$ | $0.305_{001}$ | $0.305_{001}$ |
| Average | 0.323 | 0.368 | 0.369 | 0.342 | 0.395 | 0.399 | 0.361 | 0.415 | 0.406 |

**Biomarker examples** We present two biomarker examples from a sample in the IDC dataset (Figure 7), demonstrating that STFlow accurately predicts spatially resolved gene expression with strong alignment to the ground truth.

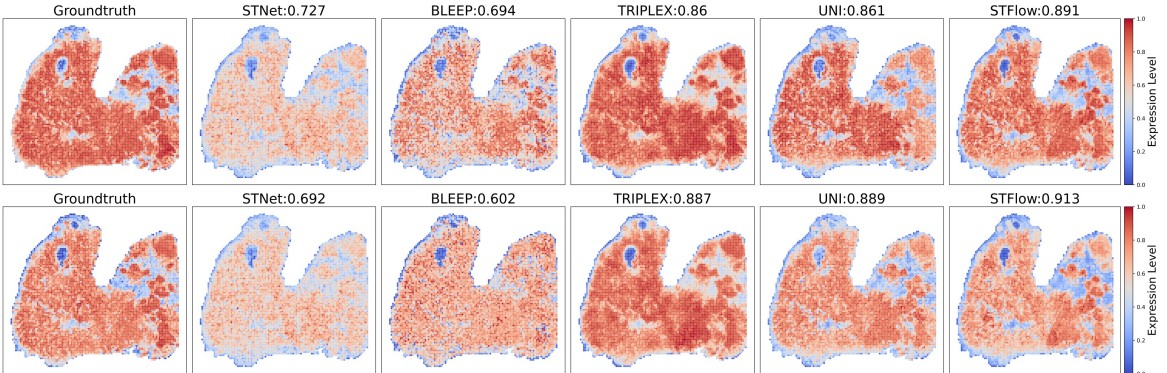

**Expression of GATA3 (top) and ERBB2 (bottom) at TENX95**

*Figure 7.* STFlow for Biomarker Discovery in Breast Samples (IDC dataset from HEST-1K).

**Hyperparameter study of ZINB distribution** We set the zero-inflation probability $\pi = 0.5$, as higher dropout rates degrade ZINB to a near-zero distribution. We conducted an ablation over the mean $\mu \in \{0.1, 0.2, 0.4\}$ and dispersion $\phi \in \{1, 2, 4\}$, with results shown in Table 13. STFlow is generally robust to these hyperparameters due to its iterative refinement, though performance can vary slightly on some datasets (e.g., 0.496–0.507 on PAAD).

*Table 13.* Hyperparameter study on ZINB distribution.

|  | (0.1,1) | (0.2,1) | (0.4,1) | (0.1,2) | (0.2,2) | (0.4,2) | (0.1,4) | (0.2,4) | (0.4,4) |
|---|---|---|---|---|---|---|---|---|---|
| IDC | $0.586_{001}$ | $0.585_{003}$ | $0.585_{001}$ | $0.584_{001}$ | $0.585_{002}$ | $0.587_{003}$ | $0.585_{001}$ | $0.583_{001}$ | $0.585_{001}$ |
| PRAD | $0.417_{002}$ | $0.415_{001}$ | $0.413_{000}$ | $0.415_{000}$ | $0.421_{002}$ | $0.413_{001}$ | $0.415_{000}$ | $0.415_{002}$ | $0.418_{001}$ |
| PAAD | $0.496_{002}$ | $0.507_{004}$ | $0.499_{000}$ | $0.499_{005}$ | $0.502_{004}$ | $0.500_{001}$ | $0.498_{000}$ | $0.499_{002}$ | $0.502_{006}$ |
| SKCM | $0.707_{007}$ | $0.704_{002}$ | $0.710_{004}$ | $0.704_{007}$ | $0.709_{003}$ | $0.704_{009}$ | $0.709_{003}$ | $0.703_{005}$ | $0.707_{008}$ |
| COAD | $0.339_{002}$ | $0.342_{003}$ | $0.341_{004}$ | $0.343_{000}$ | $0.338_{003}$ | $0.343_{004}$ | $0.342_{002}$ | $0.343_{006}$ | $0.340_{000}$ |
| READ | $0.253_{003}$ | $0.231_{000}$ | $0.243_{000}$ | $0.247_{004}$ | $0.244_{004}$ | $0.245_{000}$ | $0.236_{001}$ | $0.246_{000}$ | $0.249_{002}$ |
| CCRCC | $0.339_{001}$ | $0.337_{003}$ | $0.340_{004}$ | $0.334_{000}$ | $0.342_{008}$ | $0.329_{005}$ | $0.334_{000}$ | $0.336_{002}$ | $0.337_{003}$ |
| HCC | $0.118_{001}$ | $0.122_{000}$ | $0.123_{005}$ | $0.120_{003}$ | $0.120_{001}$ | $0.122_{004}$ | $0.126_{002}$ | $0.125_{003}$ | $0.124_{003}$ |
| LUNG | $0.611_{001}$ | $0.611_{000}$ | $0.611_{001}$ | $0.610_{000}$ | $0.611_{001}$ | $0.610_{001}$ | $0.608_{001}$ | $0.609_{000}$ | $0.613_{001}$ |
| LYMPH | $0.308_{001}$ | $0.310_{002}$ | $0.308_{001}$ | $0.309_{000}$ | $0.308_{001}$ | $0.304_{000}$ | $0.306_{001}$ | $0.305_{002}$ | $0.304_{002}$ |

