# OpenReview forum: "Scalable Generation of Spatial Transcriptomics from Histology Images via Whole-Slide Flow Matching"
_ICML.cc/2025/Conference — ICML 2025 spotlightposter_

### Official Review · Reviewer_LYQK · 2025-03-03

**Overall Recommendation:** 4

**Summary:**

This paper proposes a method to predict gene expression from histology whole-slide images using generative flow matching. Spatial transcriptomics (ST) datasets are used to train and evaluate the model. A foundation-model encoder is used to extract visual features. Spatial attention is used to model spatial dependencies. Flow matching models the joint distribution of gene expression over the whole slide in an iterative fashion.

## update after rebuttal:
After carefully reviewing the author's rebuttal and considering other reviewer's input, i decide to keep my accept rating.

**Claims And Evidence:**

The method claim to better consider cell-cell interaction through both spatial attention at the encoder and the flow matching iterative generative process. Although spatial attention clearly provides a way to model spatial dependencies, it is not very clear intuitively how the flow matching (FM) helps. Nevertheless, the ablation study shows that without FM, there's a small performance degradation on all benchmarks.

**Essential References Not Discussed:**

Not that i know of.

**Experimental Designs Or Analyses:**

The experimental section includes 8 baselines (5 spot-base and 3 slide-based) across two benchmark ST datasets (each covering several organs). Further, two ablation studies verify the effectiveness of each module of the proposed approach, showing the improvement for each one.
Overall, this is a sound and valid experimental section.

**Methods And Evaluation Criteria:**

The proposed method is appropriate for the problem. In particular, it has been proven in the litterature that ST model benefits from spatial attention as the tumor generally grows out spatially. THe evaluation benchmark dataset are also appropriate for the task and have been used previously in the literature.

**Other Comments Or Suggestions:**

no typos to report!

**Other Strengths And Weaknesses:**

strengths:
- The paper applies flow-matching methods (used in generative modeling of molecule and proteins) to gene expressions regression from image encodings. This is novel, as far as i know.
- a strong experimental analysis with extensive benchmark, ablation + complexity studies
- outperforms SOTA methods
- well written and organized. Clear math and algorithms.
- code repository is proposed

cons:
- a complex system with several modules and hyperparameters makes it challenging to evaluate
-

**Questions For Authors:**

No questions.

**Relation To Broader Scientific Literature:**

The key contributions are well-related to specific litterature, in my opinion. Both key contributions (spatial attention and flow matching) are properly related to current publications and the ST litterature is also well referenced.

**Theoretical Claims:**

No theoretical claims are made in the paper.

---

> ### Author Rebuttal · Authors · 2025-03-31
>
> We sincerely thank you for the insightful comments and will revise the manuscript accordingly. Please see our detailed responses below:
>
> ```
> A complex system with several modules and hyperparameters makes it challenging to evaluate.
> ```
>
> Ans: Thank you so much for the feedback. STFlow primarily consists of two core components: (1) a frame averaging (FA)-based encoder and (2) a flow matching (FM) optimization framework. We evaluate their effectiveness by (i) comparing different geometric encoders and ablating FA, and (ii) removing the iterative refinement process. Additionally, we provide results on varying the number of refinement steps and ZINB prior hyperparameters to further assess stability below. We will refine our discussion to more clearly highlight each component’s contribution.
>
> | #steps | S=1 | S=2 | S=5 | S=10 | S=16 |
> | --- | --- | --- | --- | --- | --- |
> | IDC | 0.580(.005) | 0.585(.002) | 0.587(.003) | 0.585(.001) | 0.585(.001) |
> | PRAD | 0.420(.003) | 0.416(.003) | 0.421(.002) | 0.414(.003) | 0.415(.004) |
> | PAAD | 0.488(.001) | 0.498(.001) | 0.507(.004) | 0.499(.001) | 0.498(.001) |
> | SKCM | 0.705(.002) | 0.707(.005) | 0.704(.005) | 0.703(.005) | 0.703(.005) |
> | COAD | 0.315(.008) | 0.343(.004) | 0.326(.009) | 0.320(.003) | 0.321(.004) |
> | READ | 0.232(.009) | 0.239(.002) | 0.240(.014) | 0.239(.003) | 0.239(.004) |
> | CCRCC | 0.322(.001) | 0.340(.002) | 0.332(.003) | 0.330(.002) | 0.319(.003) |
> | HCC | 0.115(.008) | 0.119(.002) | 0.124(.004) | 0.117(.003) | 0.118(.002) |
> | LUNG | 0.604(.002) | 0.612(.002) | 0.610(.002) | 0.611(.002) | 0.611(.001) |
> | LYMPH_IDC | 0.278(.002) | 0.310(.001) | 0.305(.001) | 0.306(.001) | 0.305(.001) |
> | Average | 0.405 | 0.417 | 0.415 | 0.412 | 0.411 |
>
> | ZINB(mean, number of failures) | (0.1,1) | (0.2,1) | (0.4,1) | (0.1,2) | (0.2,2) | (0.4,2) | (0.1,4) | (0.2,4) | (0.4,4) |
> | --- | --- | --- | --- | --- | --- | --- | --- | --- | --- |
> | IDC | 0.586(.001) | 0.585(.003) | 0.585(.001) | 0.584(.001) | 0.585(.002) | 0.587(.003) | 0.585(.001) | 0.583(.001) | 0.585(.001) |
> | PRAD | 0.417(.002) | 0.415(.001) | 0.413(.000) | 0.415(.000) | 0.421(.002) | 0.413(.001) | 0.415(.000) | 0.415(.002) | 0.418(.001) |
> | PAAD | 0.496(.002) | 0.507(.004) | 0.499(.000) | 0.499(.005) | 0.502(.004) | 0.500(.001) | 0.498(.000) | 0.499(.002) | 0.502(.006) |
> | SKCM | 0.707(.007) | 0.704(.002) | 0.710(.004) | 0.704(.007) | 0.709(.003) | 0.704(.009) | 0.709(.003) | 0.703(.005) | 0.707(.008) |
> | COAD | 0.339(.002) | 0.342(.003) | 0.341(.004) | 0.343(.000) | 0.338(.003) | 0.343(.004) | 0.342(.002) | 0.343(.006) | 0.340(.000) |
> | READ | 0.253(.003) | 0.231(.000) | 0.243(.000) | 0.247(.004) | 0.244(.004) | 0.245(.000) | 0.236(.001) | 0.246(.000) | 0.249(.002) |
> | CCRCC | 0.339(.001) | 0.337(.003) | 0.340(.004) | 0.334(.000) | 0.342(.008) | 0.329(.005) | 0.334(.000) | 0.336(.002) | 0.337(.003) |
> | HCC | 0.118(.001) | 0.122(.000) | 0.123(.005) | 0.120(.003) | 0.120(.001) | 0.122(.004) | 0.126(.002) | 0.125(.003) | 0.124(.003) |
> | LUNG | 0.611(.001) | 0.611(.000) | 0.611(.001) | 0.610(.000) | 0.611(.001) | 0.610(.001) | 0.608(.001) | 0.609(.000) | 0.613(.001) |
> | LYMPH_IDC | 0.308(.001) | 0.310(.002) | 0.308(.001) | 0.309(.000) | 0.308(.001) | 0.304(.000) | 0.306(.001) | 0.305(.002) | 0.304(.002) |

---

### Official Review · Reviewer_iKre · 2025-03-13

**Overall Recommendation:** 3

**Summary:**

This paper proposes a flow matching (FM) approach (called STFlow) for predicting the spatial transcriptomics (ST) from pathological Whole-Slide Images (WSIs). The core designs of STFlow contain i) learning the joint distribution $p(Y_0,\cdots,Y_N|I_0,\cdots,I_N)$ using the FM approach and ii) the E(2)-invariant spatial attention that adapts frame averaging (FA) to the attention operation for learning invariant spot-level representations. Two large-scale benchmarks are adopted to verify the effectiveness of STFlow. The experimental results show the superiority of STFlow over existing methods.

## update after rebuttal

Thanks for the author's rebuttal and the additional results. My concerns are well addressed. I am happy to raise my score from 2 to 3. The authors are encouraged to include the addition results into the revised paper and also carefully revised the paper to further improve the presentation quality.

**Claims And Evidence:**

Most of the claims are justified by empirical experiments. However,
- the effectiveness of the author's one core design, *i.e.*, modeling the joint distribution $p(Y_0,\cdots,Y_N|I_0,\cdots,I_N)$, seems to lack convincing evidence.
- In addition, I have several concerns about the experimental design and the results, which are specified below for the authors to answer in the rebuttal.

**Essential References Not Discussed:**

No

**Experimental Designs Or Analyses:**

Yes, I have carefully checked the experimental design and the results. I have several concerns about them:
- Lack of the experiments for the setting of $S$ and the visualization of the inference process of FM. These experiments could help to validate the stability of the proposed method and help readers better understand the model.
- Some results have large variance (Table 1). This calls into question the stability of the proposed method.

**Methods And Evaluation Criteria:**

**Methods**: Yes, the proposed methods make sense for the problem studied in this paper.

**Evaluation Criteria**: Some commonly-used metrics are not presented in the paper.

**Other Comments Or Suggestions:**

Please see *Questions For Authors*

**Other Strengths And Weaknesses:**

Strengths:
- This paper propose a new generative approach to predicting ST from WSIs. It learns the joint distribution $p(Y_0,\cdots,Y_N|I_0,\cdots,I_N)$ using the FM framework. This seems novel and could be a valid contribution to the field.
- The proposed STFlow considers to learn the invariant representation for spots via E(2)-invariant spatial attention.
- Impressive results are obtained in this paper. STFlow often outperforms existing methods by large margins.

Weaknesses:
- The presentation quality is subpar. The authors fail to clarify how they tackle the issues of existing methods in Introduction. Please see *Questions For Authors* for more details
- Some important experiments are not presented in this paper. In addition, an important design in STFlow, *i.e.*, modeling the joint distribution $p(Y_0,\cdots,Y_N|I_0,\cdots,I_N)$, seems to lack justification. Please see *Questions For Authors* for more details

**Questions For Authors:**

The proposed approach seems interesting and novel. Moreover, some promising results are obtained in this paper. Overall, I acknowledge this paper's technical contribution to the field and the novelty of the proposed scheme. However, I have several concerns about the presentation quality and experiments of this work. My concerns and questions are as follows:
- The Introduction analyzes the issues of existing schemes yet does not clearly explain how this work tackles these issues. Although some explanations are made in the following sections, this way could lead to inefficient reading.
- The layout of some Tabs and Figs could be improved, *e.g.*, Tab 2 on page 6.
- Could the authors provide the visualization of the inference process of FM. This experiments could help readers better understand STFlow.
- Could the authors represent the results of STFlow with different $S$. It may help to validate the stability of the proposed method.
- Some results have large variance (Table 1). This may call into question the stability of the proposed method.
- Some commonly-used metrics seem missing, *e.g.*, those metrics presented in the compared methods, TRIPLEX and HisToGene.
- How is the `w/o FM` implemented? I fail to find the details.
- The effectiveness of one core design, *i.e.*, modeling the joint distribution $p(Y_0,\cdots,Y_N|I_0,\cdots,I_N)$, seems to lack convincing evidence. How is this core design justified? Could the authors explain this?
- It would be better to provide an intuitive expatiation for how FM realizes the modeling of joint distribution. This could be more friendly to the reader unfamiliar with FM.
- In Algo.2, is $Y_0$ sampled once or multiple times when deriving the prediction of one WSI from the model?

If my concerns could be resolved, I would be happy to raise my rating.

**Relation To Broader Scientific Literature:**

No. The key contribution of this paper seems independent with existing works.

**Theoretical Claims:**

N/A

---

> ### Author Rebuttal · Authors · 2025-03-31
>
> We sincerely thank you for the insightful comments and will revise the manuscript accordingly. Due to character limits, all additional results are available in our anonymous codebase: https://anonymous.4open.science/r/Anonymous_STFlow-3616/.
>
> ```
> The Introduction analyzes the issues of existing schemes yet does not clearly explain how this work tackles these issues.
> ```
>
> Ans: STFlow addresses the issues as follows: (1) it reduces computational complexity via a spatial local attention mechanism, (2) it enhances spatial dependency modeling by encoding and incorporating relative orientation into attention, and (3) it captures cell-cell interactions through flow matching that effectively uses gene expression as context. We will clarify these points in the manuscript.
>
> ```
> The layout of some Tabs and Figs could be improved, *e.g.*, Tab 2 on page 6.
> ```
>
> Ans: Thank you for pointing this out! We will improve the layout of all tables and figures in the revised manuscript.
>
> ```
> Could the authors provide the visualization of the inference process of FM. This experiments could help readers better understand STFlow.
> ```
>
> Ans: We include two examples showing how STFlow refines gene expression predictions in [link](https://anonymous.4open.science/r/Anonymous_STFlow-3616/rebuttal/refinement_process.png). Step 1 shows the initial random sample Y_0, and Steps 2–5 indicate progressive denoising. The results illustrate how flow matching gradually converges to the final prediction via interpolation with a decay coefficient. The results will be included in the revised manuscript.
>
> ```
> Could the authors represent the results of STFlow with different S.
> ```
>
> Ans: We report STFlow’s performance across different numbers of refinement steps in [link](https://anonymous.4open.science/r/Anonymous_STFlow-3616/rebuttal/FM_sample_steps.md). The results show a clear improvement from one-step prediction (S=1) to introducing iterative refinement (S=2). In certain datasets, performance improves up to S = 5 (e.g., PAAD: 0.488 → 0.507). However, gains plateau or slightly decline beyond that. This observation aligns with prior works such as AlphaFlow and RNAFlow, which adopt S=5 as the default setting. The analysis will be included in the revised manuscript.
>
> ```
> Some results have large variance (Table 1).
> ```
>
> Ans: For the HEST benchmark with cross-validation, the variances are generally low, indicating stable performance. On STImage, which uses a train-val-test split, slide-based methods show higher variance than spot-based ones, likely due to variability in spatial patterns and the noise introduced by slide-level aggregation. Nevertheless, STFlow achieves lower variance than the SOTA baseline TRIPLEX in most cases.
>
> ```
> Some commonly-used metrics seem missing.
> ```
>
> Ans: We followed the HEST-1k benchmark setup, which uses Pearson correlation to evaluate performance. Metrics like MSE or MAE depend on absolute expression values and can be influenced by different normalization strategies, making direct comparisons less reliable. The MSE metrics on HEST are shown in [link](https://anonymous.4open.science/r/Anonymous_STFlow-3616/rebuttal/MSE.md), from which STFlow achieves the best performance.
>
> ```
> How is the w/o FM implemented?
> ```
>
> Ans: "STFlow w/o FM" refers to the model where the iterative refinement is removed and one-step prediction is performed, i.e., setting the number of sampling steps S = 1. We will clarify this in the revised manuscript to avoid confusion.
>
> ```
> The effectiveness of one core design, *i.e.*, modeling the joint distribution p(Y0,⋯,YN|I0,⋯,IN), seems to lack convincing evidence.
> ```
>
> Ans: Modeling the joint distribution captures dependencies across spatial spots, rather than predicting each independently. Our flow matching framework enables this by using whole-slide gene expressions as context during iterative refinement. Its effectiveness is shown in the “STFlow w/o FM” ablation, where removing the refinement process reduces the model to a one-step predictor. Prior gene imputation studies also support the value of leveraging spatial gene expression context.
>
> ```
> It would be better to provide an intuitive explanation for how FM realizes the modeling of joint distribution.
> ```
>
> Ans: Flow matching models the joint distribution by iteratively refining each spot’s gene expression using predictions from neighboring spots. This allows information to flow across spatial locations, so each prediction is influenced by its context. Through this process, the model learns how gene expressions co-vary across the tissue, effectively capturing the joint distribution.
>
> ```
> In Algo.2, is Y0 sampled once or multiple times when deriving the prediction of one WSI from the model?
> ```
>
> Ans: In our current implementation, Y_0 is sampled once per inference. A potential improvement is to sample multiple times and select the most confident prediction using a confidence model, which will be explored for future work.

---

> > ### Comment · Reviewer_iKre · 2025-04-02
> >
> > Thanks for the author's rebuttal and the additional results. The authors are encouraged to include the addition results into the revised paper and also carefully revised the paper to further improve the presentation quality.

---

> > > ### Author Response · Authors · 2025-04-02
> > >
> > > We sincerely appreciate your suggestions and will revise our paper accordingly. All results will be included in the manuscript.

---

### Official Review · Reviewer_kzNE · 2025-03-14

**Overall Recommendation:** 4

**Summary:**

The paper introduces a scalable and efficient framework for predicting spatial transcriptomics from histology images. By integrating flow matching for progressive gene refinement, E(2)-invariant spatial attention for robust spatial modeling, and whole-slide scalability, STFlow formulates gene expression prediction as a generative modeling task, effectively capturing spatial dependencies and cell-cell interactions while overcoming the limitations of existing approaches.  Experimental results demonstrate that STFlow achieves state-of-the-art performance on 17 benchmark datasets (HEST-1k and STImage-1K4M), excelling in gene expression prediction and biomarker identification tasks. It outperforms pathology foundation models in spatial gene expression prediction and achieves the highest correlation scores in biomarker gene prediction (GATA3, ERBB2, UBE2C, VWF), demonstrating its effectiveness in modeling spatial dependencies and gene interactions.

**Claims And Evidence:**

Overall, the claims in the paper are well-supported by quantitative evidence. However, to further validate the choice of ZINB, an ablation study on its hyperparameters (μ,ϕ,π) should be conducted to demonstrate their impact on prediction performance. Additionally, a hyperparameter study on the number of refinement steps in the flow matching process is needed to assess its contribution to model accuracy and convergence stability.

**Essential References Not Discussed:**

No

**Experimental Designs Or Analyses:**

The study introduces STFlow, a deep learning-based approach for predicting spatial transcriptomics (ST) data from histology images. Overall, the experimental design and analysis are rigorous, and the effectiveness of the model has been validated through multiple evaluations.
Regarding the experimental design, the study compares spot-based and slide-based methods to ensure a comprehensive performance assessment. Additionally, it employs two large-scale benchmark datasets, HEST-1k and STImage-1K4M, which help reduce dataset-specific biases. To prevent data leakage, k-fold cross-validation is used, with patient-stratified splits ensuring that training and test sets do not overlap at the patient level. The study also conducts ablation experiments to assess the contributions of key components, such as the flow matching mechanism and spatial attention module, to model performance.
In terms of analysis, the study employs Pearson correlation coefficient as the primary evaluation metric to measure the relationship between predicted and actual gene expression levels. Additionally, it evaluates model performance on four critical biomarkers (GATA3, ERBB2, UBE2C, and VWF), demonstrating superior predictive accuracy over existing methods. The study also provides visualizations of gene expression predictions, showing a strong alignment between STFlow’s predictions and ground-truth expression levels, enhancing interpretability.

**Methods And Evaluation Criteria:**

The proposed methods and evaluation criteria in the submission appear well-aligned with the problem of spatial transcriptomics prediction.
STFlow addresses the challenge of predicting spatial transcriptomics from histology images by introducing a flow matching-based generative model. STFlow explicitly models the joint distribution of gene expression across the entire slide, incorporates cell-cell interactions, and employs a local spatial attention-based slide-level encoder to reduce computational overhead. This approach overcomes the limitations of previous methods, which struggled with capturing spatial dependencies and suffered from high computational complexity.
Evaluated on two large-scale benchmark datasets, HEST-1k and STImage-1K4M, STFlow outperforms eight state-of-the-art baselines in gene expression and biomarker prediction, achieving a relative improvement compared to pathology foundation models while demonstrating superior computational efficiency.
The evaluation criteria include Pearson correlation for gene expression prediction, accuracy in predicting four key biomarkers, and computational efficiency metrics such as runtime and memory usage, ensuring a robust and comprehensive assessment.

**Other Comments Or Suggestions:**

No

**Other Strengths And Weaknesses:**

This paper is clearly written and easy to read. It presents a novel approach to spatial transcriptomics prediction by introducing flow matching-based generative modeling, which effectively models joint gene expression distributions across entire slides while incorporating cell-cell interactions. STFlow demonstrates state-of-the-art performance on HEST-1k and STImage-1K4M, achieving improvement over pathology foundation models. Additionally, it excels in biomarker prediction experiments, accurately predicting key genes such as GATA3, ERBB2, UBE2C, and VWF, highlighting its potential for clinical applications.

**Questions For Authors:**

1.	The authors should perform a hyperparameter study on the number of refinement steps in the flow matching process.
2.	The authors should conduct an ablation study on the hyperparameters of the ZINB prior (μ, ϕ, π) to demonstrate their impact on prediction performance.

**Relation To Broader Scientific Literature:**

First, Previous methods either predicted gene expression independently for each spot, neglecting cell-cell interactions, or relied on computationally expensive global attention mechanisms. STFlow overcomes these issues by introducing a flow matching-based generative modeling framework, which models the joint distribution of gene expression across an entire slide. This allows for iterative refinement, leading to more biologically meaningful predictions.
Second, STFlow leverages spatial attention with E(2)-invariant properties, ensuring robustness to spatial variations such as rotation and translation.

**Theoretical Claims:**

No

---

> ### Author Rebuttal · Authors · 2025-03-31
>
> Thanks for your feedback on our work! We will explain your concerns point by point.
>
> ```
> The authors should perform a hyperparameter study on the number of refinement steps in the flow matching process.
> ```
>
> Ans: We report STFlow’s performance across different numbers of refinement steps below. The results show a clear improvement from one-step prediction (S=1) to introducing iterative refinement (S=2). In some datasets, performance continues to improve up to S = 5 (e.g., PAAD: 0.488 → 0.507). However, gains plateau or slightly decline beyond that. This trend aligns with prior works like AlphaFlow and RNAFlow, which also adopt S=5 as default. The analysis will be included in the revised manuscript.
>
> | #steps | S=1 | S=2 | S=5 | S=10 | S=16 |
> | --- | --- | --- | --- | --- | --- |
> | IDC | 0.580(.005) | 0.585(.002) | 0.587(.003) | 0.585(.001) | 0.585(.001) |
> | PRAD | 0.420(.003) | 0.416(.003) | 0.421(.002) | 0.414(.003) | 0.415(.004) |
> | PAAD | 0.488(.001) | 0.498(.001) | 0.507(.004) | 0.499(.001) | 0.498(.001) |
> | SKCM | 0.705(.002) | 0.707(.005) | 0.704(.005) | 0.703(.005) | 0.703(.005) |
> | COAD | 0.315(.008) | 0.343(.004) | 0.326(.009) | 0.320(.003) | 0.321(.004) |
> | READ | 0.232(.009) | 0.239(.002) | 0.240(.014) | 0.239(.003) | 0.239(.004) |
> | CCRCC | 0.322(.001) | 0.340(.002) | 0.332(.003) | 0.330(.002) | 0.319(.003) |
> | HCC | 0.115(.008) | 0.119(.002) | 0.124(.004) | 0.117(.003) | 0.118(.002) |
> | LUNG | 0.604(.002) | 0.612(.002) | 0.610(.002) | 0.611(.002) | 0.611(.001) |
> | LYMPH_IDC | 0.278(.002) | 0.310(.001) | 0.305(.001) | 0.306(.001) | 0.305(.001) |
> | Average | 0.405 | 0.417 | 0.415 | 0.412 | 0.411 |
>
> ```
> The authors should conduct an ablation study on the hyperparameters of the ZINB prior (μ, ϕ, π) to demonstrate their impact on prediction performance.
> ```
>
> Ans: We set the zero-inflation probability $\pi=0.5$, as higher dropout rates degrade ZINB to a near-zero distribution. We conducted an ablation over the mean $\mu\in\{0.1,0.2,0.4\}$ and dispersion $\phi\in\{1,2,4\}$, with results shown below. STFlow is generally robust to these hyperparameters due to its iterative refinement, though performance can vary slightly on some datasets (e.g., 0.496–0.507 on PAAD). Exploring automated strategies for tuning prior parameters is a promising direction, as noted in our Limitation section.
>
> | ZINB(mean, number of failures) | (0.1,1) | (0.2,1) | (0.4,1) | (0.1,2) | (0.2,2) | (0.4,2) | (0.1,4) | (0.2,4) | (0.4,4) |
> | --- | --- | --- | --- | --- | --- | --- | --- | --- | --- |
> | IDC | 0.586(.001) | 0.585(.003) | 0.585(.001) | 0.584(.001) | 0.585(.002) | 0.587(.003) | 0.585(.001) | 0.583(.001) | 0.585(.001) |
> | PRAD | 0.417(.002) | 0.415(.001) | 0.413(.000) | 0.415(.000) | 0.421(.002) | 0.413(.001) | 0.415(.000) | 0.415(.002) | 0.418(.001) |
> | PAAD | 0.496(.002) | 0.507(.004) | 0.499(.000) | 0.499(.005) | 0.502(.004) | 0.500(.001) | 0.498(.000) | 0.499(.002) | 0.502(.006) |
> | SKCM | 0.707(.007) | 0.704(.002) | 0.710(.004) | 0.704(.007) | 0.709(.003) | 0.704(.009) | 0.709(.003) | 0.703(.005) | 0.707(.008) |
> | COAD | 0.339(.002) | 0.342(.003) | 0.341(.004) | 0.343(.000) | 0.338(.003) | 0.343(.004) | 0.342(.002) | 0.343(.006) | 0.340(.000) |
> | READ | 0.253(.003) | 0.231(.000) | 0.243(.000) | 0.247(.004) | 0.244(.004) | 0.245(.000) | 0.236(.001) | 0.246(.000) | 0.249(.002) |
> | CCRCC | 0.339(.001) | 0.337(.003) | 0.340(.004) | 0.334(.000) | 0.342(.008) | 0.329(.005) | 0.334(.000) | 0.336(.002) | 0.337(.003) |
> | HCC | 0.118(.001) | 0.122(.000) | 0.123(.005) | 0.120(.003) | 0.120(.001) | 0.122(.004) | 0.126(.002) | 0.125(.003) | 0.124(.003) |
> | LUNG | 0.611(.001) | 0.611(.000) | 0.611(.001) | 0.610(.000) | 0.611(.001) | 0.610(.001) | 0.608(.001) | 0.609(.000) | 0.613(.001) |
> | LYMPH_IDC | 0.308(.001) | 0.310(.002) | 0.308(.001) | 0.309(.000) | 0.308(.001) | 0.304(.000) | 0.306(.001) | 0.305(.002) | 0.304(.002) |

---

> > ### Comment · Reviewer_kzNE · 2025-04-03
> >
> > The author has addressed my concern.

---

> > > ### Author Response · Authors · 2025-04-03
> > >
> > > We are glad to hear that. Thanks so much for your effort in reviewing!

---

### Official Review · Reviewer_WwPE · 2025-03-14

**Overall Recommendation:** 4

**Summary:**

The authors propose STFlow, a model for spatially resolved gene-expression prediction from WSIs.

STFlow is based on flow matching, modelling the joint distribution of the full spatial gene-expression data across each WSI, through an iterative refinement process. This enables explicit modelling of spot-to-spot interactions.

The denoiser network is a frame-averaging transformer, integrating spatial context and gene interactions within the attention mechanism.

The proposed STFlow is evaluated on both the HEST-1k and STImage-1K4M datasets. It performs well compared to recent spot-based and slide-based baseline models.






##########

##########

##########

##########

Update after the rebuttal:

All reviewers are positive overall, and the authors have provided a very solid rebuttal.

This is an interesting and very solid paper, I don't really see any reason for why it shouldn't be accepted. I have increased my score to "_4: Accept_".

**Claims And Evidence:**

Yes.

**Essential References Not Discussed:**

N/A.

**Experimental Designs Or Analyses:**

Solid experimental setup.

**Methods And Evaluation Criteria:**

Yes.

**Other Comments Or Suggestions:**

Questions/Suggestions:
- It would be interesting to see the computational cost for STFlow and UNI reported in Table 1. I don't really consider added computational cost a major issue, but just to see how much the transformer model and iterative refinement slows down STFlow compared to the UNI + regression layer baseline.
- Section 4.1, _"Additionally, some ST-based approaches fail to predict significantly correlated gene expression"_: Could you clarify exactly what you refer to in Table 1 here? The results for STNet and HisToGene?
- In Section 4.2, could perhaps clarify that these 3 datasets are from HEST-1k?
- It's not entirely clear to me what "STFlow w/o FM" in Table 2, Figure 4 and Table 4 means, does this correspond to setting S = 1 in Algorithm 2?
- Figure 4 is neat, it would be interesting to see more examples like these, could you perhaps add a couple to the appendix?
- It would also be interesting to see Figure 4-like visualizations of the predicted gene-expression during the iterative refinement process? I.e., what does the initial random sample $Y_0$ look like? And how does this then evolve during the S=5 refinement steps?
- It would also be interesting to see the regression accuracy as a function of the number of refinement steps S (e.g. for S = 1, 2, 4, 8, 16), does the performance increase with more and more steps, or does it quickly plateau?
- Regarding the Limitations paragraph in Section 5: I think it would be relevant to refer to the results in Table 9 in the appendix here, or at least somewhere in the main paper? Because, I think these are encouraging results? If initializing $Y_0$ with all zeros instead of sampling from the ZINB distribution, the performance drops from 0.415 to 0.407 for UNI, which is not a lot? This would still beat all baselines in Table 1? I.e., this indicates that the model is quite robust to this choice of prior?





Minor things:
- Line 86: "This enables explicit modeling cell-cell interactions" --> "This enables explicit modeling of cell-cell interactions"?
- Line 94: "WSI collections comprising total 17 benchmark datasets" --> "WSI collections comprising a total of 17 benchmark datasets"?
- Line 83: "have enabled the detecting of RNA" --> "have enabled the detection of RNA"?
- Line 96: "et al., 2024).One concurrent work" --> "et al., 2024). One concurrent work".
- Line 97: "leverages diffusion model for" --> "leverages a diffusion model for" / "leverages diffusion models for"?
- Line 111: "modeling joint distribution" --> "modeling the joint distribution"?
- Line 120: "In this work, we repurpose" --> "In this work, we reformulate"?
- Line 142: "whole-slide images (WSIs) using an FA-based Transformer", don't need to define WSIs again here.
- Line 137: "In this study, the goal of STFlow aims to predict" --> "In this study, the goal of STFlow is to predict"?
- Line 185: "However, standard regression objective cannot model cell-cell interaction as it predicts" ---> "However, the standard regression objective cannot model cell-cell interaction as it predicts"?
- Line 194: "denoised model" --> "denoiser model"?
- "Algorithm 1 STFlow: Train" --> "Algorithm 1 STFlow: Training"?
- I think all "<--" in Algorithm 1 and 2 could be replaced with just "="?
- Line 244: "an E(2)-invariant transformation for point cloud" --> "an E(2)-invariant transformation for point clouds"?
- Line 247: "minimal modification to Transformer" --> "minimal modifications to the Transformer"?
- Line 266: "guaranteed by frame averaging framework" --> "guaranteed by the frame averaging framework"?
- Section 3.4: I think it's more common to use "Eq." instead of "Equ.".
- _Spatially Resolved Gene Expression Prediction from H&E Histology Images via Bi-modal Contrastive Learning_ is a NeurIPS 2023 paper, not 2024?
- Line 363: "Table 2 achieves the highest correlation across all biomarkers" --> "In Table 2, STFlow achieves the highest correlation across all biomarkers"?
- Line 767: "Table 10 and presents", typo.

**Other Strengths And Weaknesses:**

Strengths:
- Although the paper contains quite a few typos, it's well written overall.
- The studied problem is interesting and relevant.
- The proposed STFlow model is quite interesting, I think it makes sense to jointly model the full spatial gene-expression data using flow matching.
- The experimental evaluation is solid, utilizing both the HEST-1k and STImage-1K4M datasets.
- The proposed STFlow seems to perform well compared to relevant and recent baselines (BLEEP: NeurIPS 2023, TRIPLEX: CVPR 2024). Moreover, Section 4.3 presents relevant ablation studies which indicate that the main model components positively affect the performance.




Weaknesses:
- The paper is well written overall but does contain quite a few typos etc, it would definitely benefit from some additional careful proofreading.
- I found it quite difficult to follow and understand the method description in Section 3.3. I think Section 3.1 and 3.2 are fine, but I struggled with 3.3. I think an effort should be made to work through this section again, making sure everything is described as clearly as possible.

**Questions For Authors:**

1. Could you update Section 3.3?

2. Could you clarify what setting "STFlow w/o FM" corresponds to?

3. Could you add results for other refinement steps S (not just S = 5)?

4. Could you add some more Figure 4-like visualizations?





Justification of overall recommendation:

The studied problem is interesting and relevant, the proposed STFlow model conceptually makes sense overall, the experimental setup is solid, and STFlow seems to perform well compared to relevant baselines. While the current version requires some more proofreading and polishing, and could benefit from some additional results and visualizations (at least added to the appendix), I think that a solid rebuttal by the authors should make me want to accept this paper. I'm definitely leaning towards accept right now at least.

**Relation To Broader Scientific Literature:**

Good discussion of related work in Section 2.

**Theoretical Claims:**

N/A.

---

> ### Author Rebuttal · Authors · 2025-03-31
>
> We greatly appreciate your valuable suggestions and will revise the manuscript accordingly. Due to character limits, all additional results are available in our anonymous codebase: https://anonymous.4open.science/r/Anonymous_STFlow-3616/.
>
> ```
> I found it quite difficult to follow and understand the method description in Section 3.3
> ```
>
> Ans: We apologize for the confusion. Section 3.3 introduces the formulation of frame averaging, which may be difficult to follow due to dense notation. We will simplify the notations and streamline the equations to better highlight the core idea and improve clarity for readers.
>
> ```
> It would be interesting to see the computational cost for STFlow and UNI reported in Table 1.
> ```
>
> Ans: We present the average inference time on the test set of each dataset across splits in [link](https://anonymous.4open.science/r/Anonymous_STFlow-3616/rebuttal/time.md). The time required for visual feature extraction is excluded, as this step can be performed during preprocessing. Notably, STFlow demonstrates high efficiency due to its use of local neighborhood information.
>
> ```
> Section 4.1, *"Additionally, some ST-based approaches fail to predict significantly correlated gene expression"*: Could you clarify exactly what you refer to in Table 1 here?
> ```
>
> Ans: Yes, this refers to STNet (0.286) and HisToGene (0.237), which underperform compared to the pathology foundation model UNI (0.344) on HEST-1k. This highlights the advantage of using a foundation model and the need for an effective spatial encoder—HisToGene encodes spatial context but still underperforms. We will clarify this in the revised manuscript.
>
> ```
> In Section 4.2, could perhaps clarify that these 3 datasets are from HEST-1k?
> ```
>
> Ans: We apologize for any confusion and will clarify it in our manuscript: “We utilize two datasets from HEST and report the average correlation for each gene across different cross-validation folds. Specifically, the IDC dataset is used for GATA3 and ERBB2, while LUNG is used for UBE2C, and SKCM for VWF.”
>
> ```
> It's not entirely clear to me what "STFlow w/o FM" in Table 2, Figure 4 and Table 4 means, does this correspond to setting S = 1 in Algorithm 2?
> ```
>
> Ans: "STFlow w/o FM" refers to the model where the iterative refinement is removed and one-step prediction is performed, i.e., setting the number of sampling steps S = 1. We will clarify this in the revised manuscript to avoid confusion.
>
> ```
> Figure 4 is neat, it would be interesting to see more examples like these, could you perhaps add a couple to the appendix?
> ```
>
> Ans: We include two biomarker examples on a sample from the IDC dataset in [link](https://anonymous.4open.science/r/Anonymous_STFlow-3616/rebuttal/biomarker_case_study.png), which will be added to our manuscript.
>
> ```
> It would also be interesting to see Figure 4-like visualizations of the predicted gene-expression during the iterative refinement process?
> ```
>
> Ans: We include two examples showing how STFlow refines gene expression predictions in [link](https://anonymous.4open.science/r/Anonymous_STFlow-3616/rebuttal/refinement_process.png). Step 1 shows the initial random sample Y_0, and Steps 2–5 indicate progressive denoising. The results illustrate how flow matching gradually converges to the final prediction via interpolation with a decay coefficient. The results will be included in the revised manuscript.
>
> ```
> It would also be interesting to see the regression accuracy as a function of the number of refinement steps S.
> ```
>
> Ans: We report STFlow’s performance across different numbers of refinement steps in [link](https://anonymous.4open.science/r/Anonymous_STFlow-3616/rebuttal/FM_sample_steps.md). The results show a clear improvement from one-step prediction (S=1) to introducing iterative refinement (S=2). In some datasets, performance improves up to S = 5 (e.g., PAAD: 0.488 → 0.507). However, gains plateau or slightly decline beyond that. This trend aligns with prior works like AlphaFlow and RNAFlow, which adopt S=5 as default. The analysis will be included in the revised manuscript.
>
> ```
> Regarding the Limitations paragraph in Section 5: I think it would be relevant to refer to the results in Table 9 in the appendix here, or at least somewhere in the main paper? Because, I think these are encouraging results?
> ```
>
> Ans: Yes, the model is generally robust to the choice of prior, as the iterative refinement process helps mitigate noise from the initialization. But the ZINB distribution introduces three additional hyperparameters that can impact the performance on certain datasets, such as 0.496~0.507 on PAAD as shown in [link](https://anonymous.4open.science/r/Anonymous_STFlow-3616/rebuttal/ZINB_hyperparameters.md). Automatically estimating these parameters could be a valuable future direction.
>
> ```
> Minor things regarding the writing.
> ```
>
> Ans: We sincerely appreciate these helpful comments and will incorporate all the suggestions in the revised manuscript.

---

> > ### Comment · Reviewer_WwPE · 2025-04-06
> >
> > Thank you for the reply.
> >
> > I have read the other reviews and all rebuttals.
> >
> > All reviewers are positive overall, and the authors have provided a very solid rebuttal.
> >
> > Minor things:
> > - "Groudtruth" typo in Figure 4 and the new Figure 4-like figures shown in the rebuttal.
> > - The examples showing the iterative refinement process (https://anonymous.4open.science/r/Anonymous_STFlow-3616/rebuttal/refinement_process.png) are really neat, would be nice to see a couple of more examples like these in the appendix.
> >
> > This is an interesting and very solid paper, I don't really see any reason for why it shouldn't be accepted. I will increase my score to "4: Accept".

---

> > > ### Author Response · Authors · 2025-04-07
> > >
> > > We greatly appreciate your feedback! We will correct the typo (“Groudtruth”) and include more examples in our paper. All suggestions mentioned during the rebuttal will be incorporated into the revised manuscript.

---

### Decision · Program_Chairs · 2025-05-01

**Decision:**

Accept (spotlight poster)

**Comment:**

This paper proposes a new method to predict spatially-resolved gene expression data from H&E images at a whole slide level, through training on spatial transcriptomics data. The idea is to use generative flow matching to predict gene expressions at each location/spot/patch, and use E(2)-invariant spatial attention to aggregate representation between spatially nearby spots. The paper is generally well perceived by the reviewers for its novelty, efficacy, and thorough experimental evaluation. The rebuttal resolved some raised issues and all reviewers support the paper to be accepted.